# PLUREL: Synthetic Data unlocks Scaling Laws for Relational Foundation Models

**Vignesh Kothapalli** [1]  **Rishabh Ranjan** [1]  **Valter Hudovernik** [2]  **Vijay Prakash Dwivedi** [1]  **Johannes Hoffart** [3]
**Carlos Guestrin** [1]  **Jure Leskovec** [1]

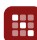 https://star-project.stanford.edu/plurel

## Abstract

Relational Foundation Models (RFMs) facilitate data-driven decision-making by learning from complex multi-table databases. However, the diverse relational databases needed to train such models are rarely public due to privacy constraints. While there are methods to generate synthetic tabular data of arbitrary size, incorporating schema structure and primary–foreign key connectivity for multi-table generation remains challenging. Here we introduce PLUREL, a framework to synthesize multi-tabular relational databases from scratch. In a step-by-step fashion, PLUREL models (1) schemas with directed graphs, (2) inter-table primary-foreign key connectivity with bipartite graphs, and, (3) feature distributions in tables via conditional causal mechanisms. The design space across these stages supports the synthesis of a wide range of diverse databases, while being computationally lightweight. Using PLUREL, we observe for the first time that (1) RFM pretraining loss exhibits power-law scaling with the number of synthetic databases and total pretraining tokens, (2) scaling the number of synthetic databases improves generalization to real databases, and (3) synthetic pretraining yields strong base models for continued pretraining on real databases. Overall, our framework and results position synthetic data scaling as a promising paradigm for RFMs.

## 1. Introduction

Large-scale publicly available pretraining data has been central to the success of Foundation Models (FMs) across text,

[1]Stanford University [2]Kumo AI [3]SAP Labs LLC. Correspondence to: <{vigneshk,ranjanr,guestrin,jure} @cs.stanford.edu>.

*Proceedings of the 43rd International Conference on Machine Learning*, Seoul, South Korea. PMLR 306, 2026. Copyright 2026 by the author(s).

image, video, speech, and other modalities (Bommasani et al., 2022; Hoffmann et al., 2022; Achiam et al., 2023; Zhou et al., 2024; Team et al., 2025; Yang et al., 2025). Similar progress has recently emerged for tabular foundation models, which demonstrate strong generalization across datasets using large-scale pretraining data (Hollmann et al., 2023; 2025; Spinaci et al., 2025; Zhang et al., 2025). However, multi-table relational databases, which constitute the primary modality for most enterprise data worldwide, remain largely inaccessible because of privacy and business constraints (Dove & Phillips, 2015; Cohen & Mello, 2018; Hoofnagle et al., 2019). This lack of public training data makes the development of RFMs challenging.

RFMs provide a novel paradigm for learning on relational databases and performing numerous predictive tasks through a single pretrained model via in-context learning. Tasks such as user churn prediction in e-commerce databases, fraud detection in financial databases, and inventory forecasting in industrial product databases can all be executed within seconds without developing individual task-specific models (Fey et al., 2025; Dwivedi et al., 2025; Ranjan et al., 2026). Just as LLMs have achieved strong performance across diverse text tasks by scaling training data to tens of trillions of tokens (Liu et al., 2025; Yang et al., 2025), RFMs may achieve similar gains with increasing data scales. Recent RFMs, despite showing promising capabilities such as zero-shot predictions (Ranjan et al., 2026; Wang et al., 2025), are trained on only a few publicly available databases and this lack of diversity hinders the benefits of further data scaling. Thus, there is a pressing need to address the lack of diverse, large-scale databases that can facilitate the development of next-generation RFMs.

Single-table models address this problem with synthetic table generation techniques, primarily using Structural Causal Models (SCMs) (Hollmann et al., 2023; 2025; Grinsztajn et al., 2025). However, a collection of isolated tables cannot sufficiently model the complexities of real-world databases (Kent, 1981), as they omit the primary-foreign key relationships between rows across different tables. Such connectivity is crucial as it determines the locality of infor-

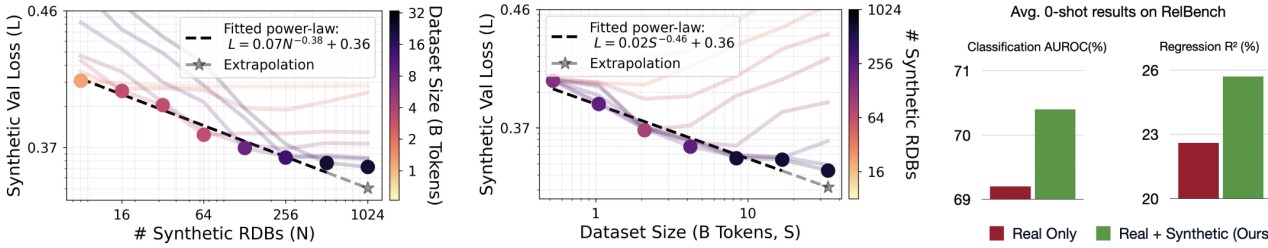

*Figure 1.* **(Left)** Pretraining loss $L$ scales as a power law with both (1) the number of synthetic databases $N$ and (2) the pretraining dataset size $S$, when not bottle-necked by the other. See Section 3.1 for details. **(Right)** On real-world predictive tasks, PLUREL-based synthetic pretraining followed by continued pretraining on real data outperforms real data pretraining alone. See Section 3.3 for details.

mation at multiple levels (i.e., at tabular and row levels) and shapes the joint data distributions that RFMs are intended to learn. The main difficulty in extending SCMs to relational data lies in incorporating the row-level primary-foreign key connectivity with the table-specific SCM mechanisms. Recent work by Hoppe et al. (2025) couples multiple SCMs through a common node for relational data generation. However, this simplifies the process into a single SCM-based data generation and fails to model the primary-foreign key connectivity. Alternative approaches such as the Synthetic Data Vault (Patki et al., 2016), GAN-based (Gueye et al., 2023), and diffusion models (Pang et al., 2024; Hudovernik, 2024; Ketata et al., 2025) can capture characteristics of real-world databases, but cannot generate novel ones from scratch without relying on existing real-world examples.

To address these limitations, we introduce PLUREL[1] , a lightweight framework for synthesizing relational databases from scratch that captures the multi-scale structural properties essential for training RFMs. We develop PLUREL through three levels of abstraction. (1) At the schema level, we design tables and their directed relationships to establish the database structure. (2) At the connectivity level, we model the bipartite relationships between tables linked via primary–foreign (P→F) relationships to populate the foreign key columns. (3) At the feature level, we employ Structural Causal Models (SCMs) combined with a conditional table generation process to incorporate temporal patterns and generate table rows. We formalize PLUREL in its most general form and demonstrate its effectiveness by pretraining Relational Transformer (RT) (Ranjan et al., 2026) models on billions of tokens from PLUREL-generated synthetic data.

By removing any data bottlenecks, PLUREL allows us to conduct scaling analyzes with respect to the number of synthetic databases (diversity) and total pretraining tokens (size). We observe power law scaling (Figure 1) finding that RT's performance improves predictably with both axes. Further, the scaling improvements show consistent zero-shot

transfer to real-world datasets, as demonstrated by forecasting tasks on unseen RelBench (Robinson et al., 2024) datasets. Synthetic pretraining synergizes well with continued pretraining on real data, showing up to $+7.4\%$ and $+5.2\%$ absolute improvements on classification AUROC and regression $R^2$ respectively.

## 2. Synthetic Relational Data Generation

We introduce the PLUREL framework through a concrete real-world example. Consider a *relational database (RDB)* in the e-commerce domain with entity tables such as Users and Items, along with activity tables such as Transactions. The database *schema* captures directed relationships between tables, such as linking Items to Transactions through a foreign key. The causal mechanisms generating the rows in this e-commerce RDB are driven by human behavior and external events over time. For instance, increased demand for winter clothing during a Black Friday sale manifests as a surge in sweater purchases. Such events induce many primary-foreign key links (P→F) from a single sweater row in Items (P) to multiple purchase rows in Transactions (F). Through these cross-table links, the database jointly captures attributes of entities (e.g., item price, user age) and activities (e.g., purchase time, quantity), distributing information across connected tables rather than isolating it within a single table.

In PLUREL, we generate synthetic databases by leveraging the abstractions mentioned above in three stages: (i) a schema is represented as a directed graph $\mathcal{G}$, where nodes correspond to a set of tables $\mathcal{T}$ and edges represent inter-table connectivity, (ii) event-driven dynamics are modeled through P→F bipartite connectivity between rows across tables, (iii) diverse attributes and joint data distributions are captured using Structural Causal Models (SCMs) when generating table rows. See Figure 2 for an overview.

### 2.1. Schema Generation via Directed Graphs

The schema determines the number of foreign key columns in each table and thereby controls information locality at

---

[1]*Plurel* is an archaic form of the word *plural*, meaning "more than one". In this paper, PLUREL refers to generating "more than one" (possibly even an unlimited number) of relational databases.

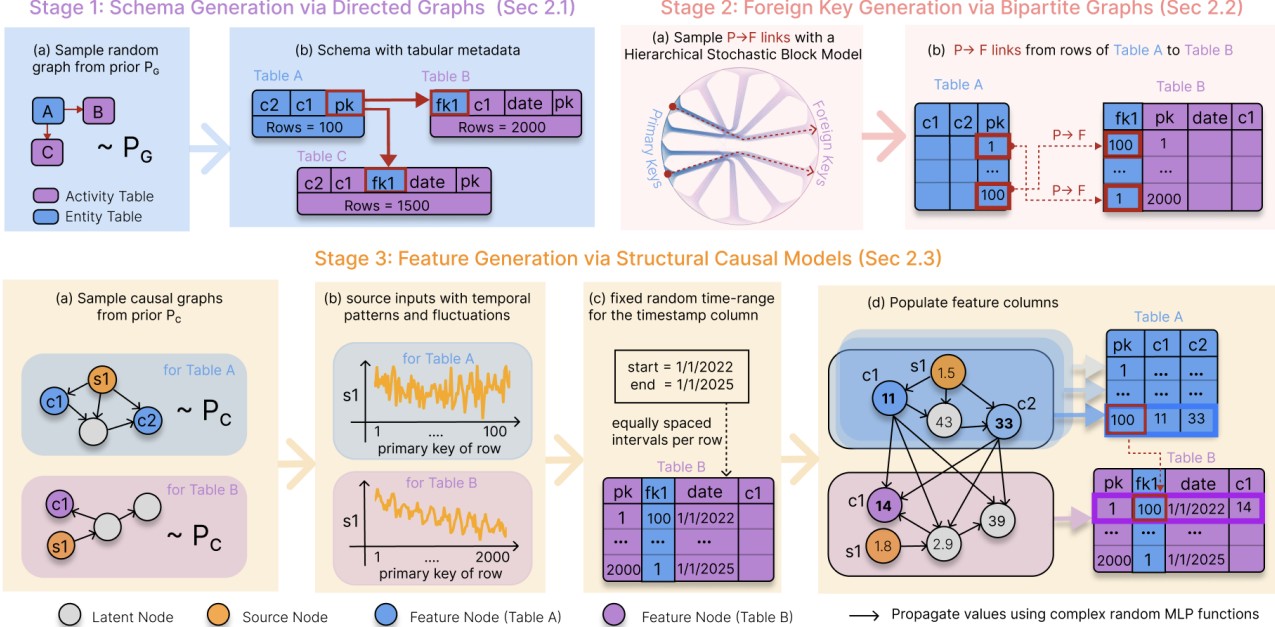

*Figure 2.* The **PLUREL** framework. **Stage 1** generates a schema by sampling a directed graph $\mathcal{G}$ and populating the metadata with row and column counts. In **Stage 2**, the foreign key columns are populated using a bipartite graph between rows of parent–child table pairs, each edge representing a primary–foreign key (P→F) link. In **Stage 3**, we follow a topological ordering of tables in $\mathcal{G}$ and leverage Structural Causal Models (SCMs) conditioned on parent tables, with temporal patterns in source node inputs to populate the feature columns.

the tabular level of an RDB. We sample $\mathcal{G}$ from a family of random directed acyclic graphs (DAGs) $\mathcal{P}_G$. We do not support cycles, which is a limitation (Appendix A). A topological ordering of $\mathcal{G}$ specifies the table generation order: tables at the first level are synthesized independently, while tables at subsequent levels are generated conditionally on the feature columns of their parent tables through P→F links. Based on their connectivity patterns in $\mathcal{G}$, we further partition tables into two categories. *Entity tables* correspond to nodes with out-degree at least one, while the remaining nodes are treated as *activity tables*. The number of rows and feature columns for each table is sampled independently from a distribution of values. Together, these design choices define the top-level schema configuration, including the number of tables, their directed relationships, table types, and associated metadata such as row and column counts.

## 2.2. Foreign Key Generation via Bipartite Graphs

Once the schema is established by $\mathcal{G}$, we move to the next stage and design the bipartite row-level connectivity between pairs of tables. Each table $T \in \mathcal{T}$ is characterized by a set of *feature columns*, a *primary key* column, and a set of (optional) *foreign key* columns. A parent table of $T$ is a predecessor of node $T$ in $\mathcal{G}$, denoted as $\widetilde{T} \in \text{Pr}(T, \mathcal{G})$. The primary key indexes the structured information within a row of table $T$, while the foreign key references a row in a parent table $\widetilde{T}$. Given a fixed number of rows per table, we treat row indices as primary key values for simplicity.

This formulation allows the foreign key column of $T$ to be populated by sampling primary keys from $\widetilde{T}$ (see Stage 2 in Figure 2). Recent works (Hudovernik et al., 2025) have shown that real-world databases exhibit a hierarchical primary–foreign key connectivity pattern between pairs of tables. Motivated by this observation, we adopt a clustering-based strategy to populate foreign key columns and control row-level information locality in an RDB. In particular, we cluster the rows of $T, \widetilde{T}$ into blocks and employ a Hierarchical Stochastic Block Model (HSBM) (Peixoto, 2014) to determine the bipartite connectivity between the rows. We repeat this procedure for all table pairs $(T, \widetilde{T})$ in the RDB.

**HSBM based connectivity.** Without loss of generality, let a table $T$ contain $N$ primary keys (rows) and one of its parent tables $\widetilde{T}$ contain $M$ primary keys. We partition these IDs into a hierarchical collection of blocks. A hierarchy $\mathbf{H}_T = (B_T^1, \dots, B_T^L)$ for table $T$ is defined by the number of levels $L$ and the number of blocks $(B_T^l)$ at each level $l \in [L] = \{1, \dots, L\}$. For example, $\mathbf{H}_T = (3, 6)$ specifies two-levels with $B_T^1 = 3$ blocks at level 1 and $B_T^2 = 6$ blocks at level 2. Using hierarchies $\mathbf{H}_T$ and $\mathbf{H}_{\widetilde{T}}$ with the same number of levels $L$, we control row-level connectivity from $\widetilde{T}$ to $T$ via level-wise probabilities $\mathbf{P}[l], l \in [L]$, as:

$$\mathbf{P}[l] = \begin{bmatrix} p_{1,1} & \cdots & p_{1,B_T^l} \\ \vdots & \ddots & \vdots \\ p_{B_{\widetilde{T}}^l,1} & \cdots & p_{B_{\widetilde{T}}^l,B_T^l} \end{bmatrix}. \qquad (1)$$

Let row $i$ in table $T$ be assigned a level-wise block vector $\mathbf{b}_i = (b_i^1, \ldots, b_i^L)$, where $b_i^l \in [B_T^l]$. Similarly, let row $j$ in table $\widetilde{T}$ be assigned $\widetilde{\mathbf{b}}_j = (\widetilde{b}_j^1, \ldots, \widetilde{b}_j^L)$, where $\widetilde{b}_j^l \in [B_{\widetilde{T}}^l]$. The probability that row $j$ of $\widetilde{T}$ links to row $i$ of $T$ is:

$$\mathbb{P}(j \to i) = \frac{s_{ij}}{\sum_{k=1}^M s_{ik}}, \qquad s_{ij} := \prod_{l=1}^L \mathbf{P}[l][\widetilde{b}_j^l, b_i^l]. \quad (2)$$

**Remark.** In the above formulation, if one sets $\mathbf{H}_T = (1)$, $\mathbf{H}_{\widetilde{T}} = (1)$ and $\mathbf{P}[1] = [1]$, then all primary keys of the parent table $\widetilde{T}$ are equally likely of being used as foreign keys in table $T$. This is a setting in which row generation for $T$ depends uniformly on all the rows of $\widetilde{T}$. The flexibility in the design of $\mathbf{P}$ thus allows rows of $T$ to depend either on many parent rows in $\widetilde{T}$ or on a small subset.

## 2.3. Feature Generation via Structural Causal Models

In the final stage, we leverage Structural Causal Models (SCMs) (Pearl, 2009; Hollmann et al., 2023; 2025) to generate the cell values in tables and complete the synthesis. We associate each table $T \in \mathcal{T}$ with its own SCM (see Stage 3 in Figure 2). An SCM is defined by a causal graph $\mathcal{C}_T = (\mathcal{V}_T, \mathcal{E}_T)$ sampled from a prior $\mathcal{P}_C$, where nodes represent variables and directed edges encode cause-and-effect relationships among them. Each node is associated with a mechanism $z_i = H_i(\text{Pr}(v_i, \mathcal{C}_T), \mathbf{u}_i)$, where $\text{Pr}(v_i, \mathcal{C}_T)$ are the predecessors of node $v_i \in \mathcal{V}$, $\mathbf{u}_i$ is an exogenous input representing latent factors not explicitly modeled in the causal graph, and $H_i$ is a deterministic (non-linear) function. The feature columns of $T$ are represented by a subset of nodes $\mathcal{V}_T^F \subseteq \mathcal{V}_T$ in the causal graph $\mathcal{C}_T$ of the SCM. The nodes without incoming edges are treated as source nodes $\mathcal{V}_T^S \subset \mathcal{V}_T$. A *realization* of an SCM corresponds to one forward pass through $\mathcal{C}_T$ with fixed exogenous inputs.

**Conditional row generation.** The tabular data synthesis follows the topological sort ordering of $\mathcal{G}$, and ensures that all the parent tables of $T$ have been synthesized before it. The first generation of tables in the topological sort of $\mathcal{G}$ will not have foreign key columns. For such $T$, we obtain the cells of a single row by (1) initializing source nodes $\mathcal{V}_T^S$, (2) propagating their values through the causal graph $\mathcal{C}_T$, and (3) collecting the values at feature nodes $\mathcal{V}_T^F$. In cases where $T$ has foreign key columns, the feature nodes $\mathcal{V}_{\widetilde{T}}^F$ of SCMs associated with all of its parent tables $\widetilde{T} \in \text{Pr}(T, \mathcal{G})$ are also considered. Formally, $z_i$ can be generalized as follows:

$$z_i = H_i\left(\cup_{\widetilde{T} \in \text{Pr}(T, \mathcal{G})} \mathcal{V}_{\widetilde{T}}^F, \text{Pr}(v_i, \mathcal{C}_T), \mathbf{u}_i\right). \quad (3)$$

When $T$ does not have foreign-key columns, then the node set represented by $\bigcup_{\widetilde{T} \in \text{Pr}(T, \mathcal{G})} \mathcal{V}_{\widetilde{T}}^F$ is empty ($\emptyset$) and $z_i$ in Equation (3) specializes to the simpler formulation above.

**Data types.** Feature columns in tables span multiple data types, including numeric, categorical, and boolean attributes.

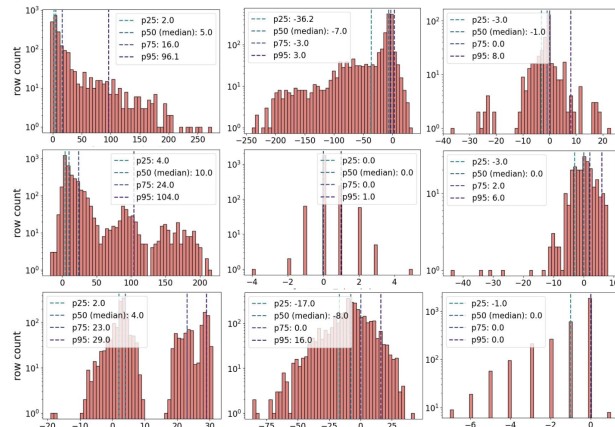

*Figure 3.* Synthesizing RDBs with PLUREL results in diverse data distributions across feature column values.

To capture this diversity, we associate each node in an SCM with either a *numeric* or *categorical* type with equal probability, enabling the construction of data-type-aware causal mechanisms. In practice, real-world databases often contain multimodal and semi-structured fields, including text, images, audio, geospatial attributes, JSON/XML objects, as well as hashed, tokenized, or encrypted columns. While our current implementation focuses on numeric and categorical features, the framework can naturally extend to these richer data modalities by augmenting the SCM mechanisms.

### 2.3.1. MODELING TEMPORAL PATTERNS

Features in real-world databases often exhibit correlations across rows due to temporally related events. We incorporate such temporal correlations across rows in PLUREL by relying on the exogenous inputs $\mathbf{u}_i$ of source nodes. We do so by modeling $\mathbf{u}_i^{(r)}$ for a row with index/primary-key $(r)$ as a combination of trend, cyclical, and fluctuation components. Furthemore, this design avoids the unrealistic assumption that features associated with identical foreign keys are independent and identically distributed (i.i.d.).

**Definition 2.1.** *The* trend : $\mathbb{R} \to \mathbb{R}$ *is a power-law function with exponent* $\alpha \in \mathbb{R}$, *a scale parameter* $s \in \mathbb{R}$, *an offset* $o \in \mathbb{R}$, *an upper-bound* $b \in \mathbb{R}$, *and total row count* $R \in \mathbb{R}$ *as:* trend$(r) = \min\left(s * \left(\frac{r}{R}\right)^\alpha + o, b\right)$.

**Definition 2.2.** *The* cycle : $\mathbb{R} \to \mathbb{R}$ *is defined by the periodicity* $p \in \mathbb{R}$, *a scale parameter* $s \in \mathbb{R}$, *a lower-bound* $l \in \mathbb{R}$, *and an upper-bound* $b \in \mathbb{R}$ *as:* cycle$(r) = \min\left(\max\left(s * \sin\left(\frac{\pi r}{p}\right), l\right), b\right)$.

**Definition 2.3.** *The* fluc : $\mathbb{R} \to \mathbb{R}$ *is defined by a random variable sampled i.i.d from the normal distribution* $n \sim N(0, 1) \in \mathbb{R}$, *a lower-bound* $l \in \mathbb{R}$, *an upper-bound* $b \in \mathbb{R}$, *and a fluctuation scale* $\lambda_n \in \mathbb{R}$ *as:* fluc$(r) = \min\left(\max\left(\lambda_n * n, l\right), b\right)$.

**Numerical inputs.** Let $g(r)$ denote the average of the *trend*,

*cycle*, and *fluc* functions for each row $r$:

$$g(r) = \mathrm{avg}\left(\mathrm{trend}(r), \mathrm{cycle}(r), \mathrm{fluc}(r)\right). \quad (4)$$

For numerical source nodes $\mathcal{V}^S$, we set $\mathbf{u}_i^{(r)} = g(r)$, and employ exogenous inputs that exhibit constant, linear, sub-linear, and super-linear trends, along with cyclical patterns of varying periodicity and bounded fluctuations.

**Categorical inputs.** For source nodes associated with the categorical type, we restrict values to the set $\{1, \ldots, C\}$. The choice of $C$ is sampled independently for each categorical node. To extend temporal structure to this setting, we associate each category $c \in [C]$ with its own numerical temporal function $g_c(r)$. For each row $r$, we then sample $\mathbf{u}_i^{(r)} \sim \mathrm{Categorical}(\mathbf{p}(r))$, where $\mathbf{p}(r) = \mathrm{Softmax}(\mathbf{g}(r))$, and $\mathbf{g}(r) = (g_1(r), \ldots, g_C(r))$. This design allows arbitrary temporal resolutions, from seconds to centuries, to be associated with the rows of tables. We represent such time ranges using the timestamp column in activity tables, thus incorporating temporal data types in synthetic RDBs.

### 2.3.2. SCM MECHANISMS

For every SCM mechanism $z_i$ associated with table $T$, the data types of the nodes inform the design of $H_i$ in Equation (3). In particular, $H_i$ follows a projection–reconstruction design. First, the values of the predecessor nodes of the same SCM ($\mathrm{Pr}(v_i, \mathcal{C}_T)$) as well as realizations of feature nodes in the parent SCM $\left(\bigcup_{\widetilde{T} \in \mathrm{Pr}(T, \mathcal{G})} \mathcal{V}_{\widetilde{T}}^F\right)$ are projected into a shared latent space. These representations are aggregated and mapped back to the $v_i$ node's data type.

**Projecting nodes.** Let $v_j \in \mathrm{Pr}(v_i, \mathcal{C}_T)$ denote a predecessor of node $v_i$ in the causal graph $\mathcal{C}_T$. If $v_j$ contains a numeric value, a randomly initialized MLP projects the value from $\mathbb{R}$ into a $d_{\mathrm{hid}}$-dimensional latent space $\mathbb{R}^{d_{\mathrm{hid}}}$. If $v_j$ contains a categorical value ($c \in [C]$), we first select the $c^{\mathrm{th}}$ row of a randomly initialized embedding matrix $\mathbf{E}_{\mathrm{proj}}^{v_j} \in \mathbb{R}^{C \times d_{\mathrm{hid}}}$ and then transform it using an MLP to obtain a latent representation in $\mathbb{R}^{d_{\mathrm{hid}}}$. The same procedure is applied to the feature nodes of the parent table's SCM realizations. Following Equation (3), this projection step is applied to all SCM nodes in $\bigcup_{\widetilde{T} \in \mathrm{Pr}(T, \mathcal{G})} \mathcal{V}_{\widetilde{T}}^F$ and $\mathrm{Pr}(v_i, \mathcal{C}_T)$.

**Reconstructing nodes.** For notational simplicity, we denote the unified set of relevant SCM nodes $\bigcup_{\widetilde{T} \in \mathrm{Pr}(T, \mathcal{G})} \mathcal{V}_{\widetilde{T}}^F$ and $\mathrm{Pr}(v_i, \mathcal{C}_T)$ by $\mathcal{M}(i)$. The exogenous input $\mathbf{u}_i \in \mathbb{R}^{d_{\mathrm{hid}}}$ for such nodes is sampled from a distribution $\xi_i$ and combined with the projected vectors $\mathbf{e}_k \in \mathbb{R}^{d_{\mathrm{hid}}}, k \in \{1, \cdots, |\mathcal{M}(i)|\}$ to form a weighted aggregate latent vector:

$$\mathbf{e}_i = w_u \mathbf{u}_i + \sum_{k=1}^{|\mathcal{M}(i)|} w_k \mathbf{e}_k. \quad (5)$$

Here $w_u \in \mathbb{R}$ controls the influence of the exogenous input, while $w_k \in \mathbb{R}$ controls the contribution of projected parent nodes. If node $v_i$ is assigned a numeric type, the aggregated representation $\mathbf{e}_i \in \mathbb{R}^{d_{\mathrm{hid}}}$ is reconstructed into $\mathbb{R}$ using a randomly initialized MLP. If $v_i$ is assigned a categorical type, $\mathbf{e}_i$ is first transformed by an MLP to obtain $\mathbf{e}_i' \in \mathbb{R}^{d_{\mathrm{hid}}}$ and then mapped to a discrete category using a randomly initialized embedding matrix $\mathbf{E}_{\mathrm{rec}}^{v_i} \in \mathbb{R}^{C \times d_{\mathrm{hid}}}$ via $\arg\max(\mathbf{E}_{\mathrm{rec}}^{v_i} \mathbf{e}_i')$. The reconstructed values of feature nodes $\mathcal{V}_T^F$ are written to their corresponding table cells in $T$.

**Summary of synthesis.** A single SCM realization generates the cell values for one row of table $T$. Repeating this execution for all the rows completes the table generation process. Extending this to all tables based on $\mathcal{G}$ synthesizes the entire RDB. As real-world RDBs tend to miss cell values due to various data collection errors, we also implant NULL values in randomly selected cells of feature columns.

## 3. Experiments

We pretrain the Relational Transformer (RT) on billions of synthetic tokens to study data scaling behavior. We focus on how PLUREL generated synthetic data diversity (number of RDBs) and dataset size (token count) affect pretraining loss and zero-shot generalization. We report scaling trends, zero-shot results on real-world tasks from RelBench (Robinson et al., 2024), and the benefit of synthetic pretraining for continued pretraining on low-diversity real-world data.

**RelBench Datasets.** We use the following 6 datasets from RelBench: `rel-amazon`, `rel-avito`, `rel-f1`, `rel-hm`, `rel-stack` and `rel-trial` as our real-world data. Each dataset comprises the relational database and the forecasting task tables. The task tables are curated using manually designed SQL operations on the database tables.

**Synthetic Datasets.** PLUREL employs a distribution of hyperparameters for synthesizing RDBs. For example: $\mathcal{G}$ is sampled from a prior of `Barabasi-Albert` (Barabási & Albert, 1999), `Reverse Random-Tree` (Prufer, 1918), and `Watts-Strogatz` (Watts & Strogatz, 1998) random graphs. These priors model a variety of table relationships with the presence of hub tables, a strictly hierarchical schema, and tight local clustering. For the MLPs used to project (or reconstruct) node values in SCM mechanisms, the activations are sampled uniformly from $\{$`relu`, `elu`, `silu`, `softsign`, `tanh`$\}$. The complete list is presented in Table 2. The synthesis of a single RDB is thus controlled only by a seed parameter and results in diverse distributions across feature columns (see Figure 3).

**Masked token prediction (MTP) and autocomplete tasks.** RT treats each table cell as a token and is pretrained using the *masked token prediction* (MTP) objective over numeric and boolean feature cells. For each masked cell, the input

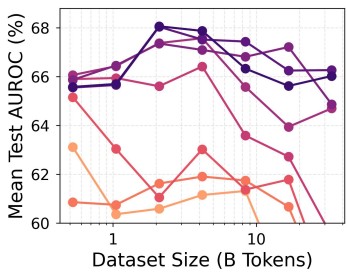 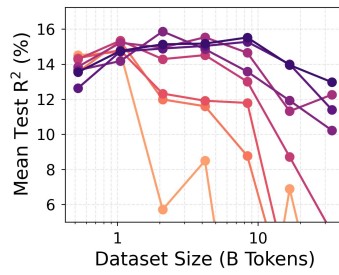 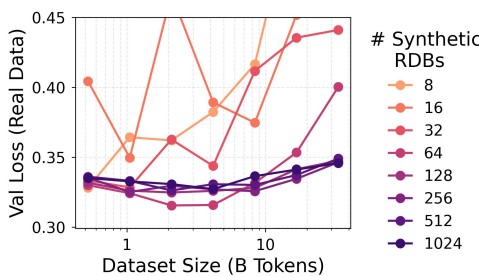

*(a)* Mean 0-shot test AUROC (%) (↑)     *(b)* Mean 0-shot test $R^2$ (%) (↑)     *(c)* Validation loss (↓) on real data (RelBench)

*Figure 4.* Validation loss and zero-shot performance on RelBench tasks. The synthetic pretraining dataset sizes (in billions of tokens) are varied along with the number of PLUREL RDBs to obtain the scaling curves. (↓)/(↑) indicates that lower/higher values are better.

context is constructed from cells in the same row and column, as well as neighboring rows connected through P→F and F→P links. We use Huber loss for numeric targets and CrossEntropy loss for boolean targets. In RelBench, autocomplete tasks mask cells in existing tables to evaluate property prediction, while forecasting tasks mask cells in curated task tables to predict future outcomes. For example, masking cells in the `item-churn` table of `rel-amazon` trains the model to predict whether a product will receive reviews in the next three months. Since PLUREL does not rely on curated task tables, masking cells in the synthetic tables naturally mirrors both property prediction and forecasting.

**Architecture and dowstream evaluation.** We use the 12 layer RT architecture as proposed by Ranjan et al. (2026) and make the following changes. (1) We do not use the 'full' attention mask, considering its limited utility, and reduce the compute overhead (Appendix C). (2) We incorporate Query-Key Normalization to the relational attention layers to stabilize training and avoid early overfitting (Appendix D.2). We measure the zero-shot performance of RT on RelBench through the 10 binary classification tasks using AUROC, and the 8 regression tasks using $R^2$ score.

**Hyperparameters and compute resources.** We use a batch size of 128, context length of 1024, BFS sampling width of 128, and use the AdamW optimizer with weight decay 0.1, a peak learning rate of $5 \times 10^{-4}$, with a linear warmup ratio of 0.2 and a linear decay to zero for the remaining steps. Experiments are conducted on 1 Blackwell B200 GPU, where one pretraining run takes around 3 hours.

### 3.1. Scaling Laws for Data Diversity and Size

We consider two axes of data scaling: (1) $N$: the number of synthetic RDBs (diversity), (2) $S$: pretraining tokens extracted from those RDBs (size). The validation loss $L$ is the mean of CrossEntropy loss for classification and Huber loss for regression over held-out synthetic RDBs. Fixing the pretraining hyperparameters as above and marginalizing out randomness from training and synthetic data generation, the

validation loss $L(N, S)$ of the final checkpoint is a function of both $N$ and $S$. Further, we define:

$$L(N) = \min_S L(N, S) \text{ and } L(S) = \min_N L(N, S).$$

We hypothesize that the loss has a power law dependency on diversity $N$ when not bottle-necked by size $S$, and similarly on size $S$ when not bottlenecked by diversity $N$. Formally, with $A_{N/S}, \alpha_{N/S}, C_{N/S} \in \mathbb{R}$ to be fit on the data:

$$L(N) = A_N N^{-\alpha_N} + C_N \quad \text{(Diversity power law)} \quad (6)$$

$$L(S) = A_S S^{-\alpha_S} + C_S \quad\quad\quad \text{(Size power law)} \quad (7)$$

To fit the 6 power law parameters, we perform a separate synthetic pretraining run for every combination in the grid $(N, S) \in \{8, 16, 32, 64, 128, 256, 512, 1024\} \times \{0.5B, 1B, 2B, 4B, 8B, 16B, 32B\}$. We measure the mean loss $L(N, S)$ of the final checkpoint on a held-out set of 10k contexts (5k each for zero-shot classification and regression) in total from 100 held-out synthetic RDBs. We compute $L(N)$ and $L(S)$ by taking the minimum loss values from this grid, and fit the parameters with the curve fitting procedure from Kaplan et al. (2020) keeping $N = 1024$ and $S = 32B$ points held-out. Thus, we fit 3 parameters $(A_N, \alpha_N, C_N)$ on 7 $(N, L(N))$ points, and the other 3 parameters $(A_S, \alpha_S, C_S)$ on 6 $(S, L(S))$ points. Finally, we check the predictive power of our scaling laws on the held-out points for $N = 1024$ and $S = 32B$. Figure 1 shows the scaling curves, including the fitted parameters.

**Observations.** We see that points on the scaling frontier roughly lie on the fitted line in the log-log plot between excess loss and $N$ or $S$, validating our power law hypothesis. Further, the extrapolated line makes a reasonable prediction at $2\times$ the data scale. We also note that to obtain the best loss, both $N$ and $S$ need to be scaled in tandem, as scaling $N$ for fixed $S$, or scaling $S$ for fixed $N$, both result in non-monotonic curves as shown by the faded lines in Figure 1.

**Remark.** We note that a joint power-law of the form

$$L(N, S) = A_N N^{-\alpha_N} + A_S S^{-\alpha_S} + C \quad (8)$$

as used by Hoffmann et al. (2022) and Ma et al. (2025) is not suitable in our case, as $L$ is not monotonic in $N$ or $S$. This can be seen in the U-shaped faded curves in Figure 1, which correspond to $L(N)$ and $L(S)$ for different values of $S$ and $N$ respectively. Intuitively, increasing diversity $N$ for fixed size $S$ leads to underfitting, and increasing size $S$ for fixed diversity $N$ leads to overfitting.

### 3.2. Generalization to Real Datasets

The masked token prediction (MTP) tasks on synthetic RDBs promote broad relational understanding in RFMs, enabling generalization beyond synthetic database specific patterns to unobserved databases. We demonstrate this behavior by computing the MTP loss on the validation split of all the 18 RelBench tasks under the same synthetic scaling setup as Section 3.1. Figure 4c shows that a lack of diversity with a smaller number of synthetic RDBs results in undesirable scaling curves for RelBench tasks. Especially, for the $\{8, 16, 32\}$ settings, the larger datasets tend to be suboptimal as the loss curves exhibit a clear upward trend. However, such behavior is mitigated as the number increases, and the benefits from scaling the dataset size become evident. Nevertheless, this is saturation of the loss as RelBench is out-of-distribution for our synthetic data. Measuring the AUROC (Figure 4a) and $R^2$ (Figure 4b) on the test splits of RelBench tasks results in similar observations, where a larger number of synthetic RDBs coupled with larger datasets can improve the overall performance.

### 3.3. Continued Pretraining on Real Datasets

Synthetic pretraining yields strong base RT models for downstream prediction and continued real-data pretraining. To pretrain on RelBench databases, we follow the *leave-one-DB-out* (Ranjan et al., 2026) approach for randomly initialized and the synthetic pretrained RT model. Specifically, the model is pretrained six times, each time holding out one RelBench dataset for evaluation, while forecasting and autocomplete tasks from the remaining five datasets are used for MTP-based pretraining. During evaluation, we select the checkpoint with the highest score on the validation split (per task) and report its score on the corresponding test split. We repeat experiments with 3 different seeds to report the mean and standard error of the metrics per task.

**Model selection.** As base model for continued pretraining, we chose the model pretrained on 1024 synthetic RDBs and 4B tokens as it maximizes the worse validation metric out of $R^2$ and AUROC without continued pretraining. Results are robust to base models, and sometimes even better for models worse on this metric (App. D.1), indicating post-hoc reversal from continued pretraining (Ranjan et al., 2024).

**Observations.** Table 1 shows that synthetic pretraining consistently improves zero-shot performance when com-

| Dataset | Task | Real only | Synthetic + Real (ours) | Absolute Gain (%) | Synthetic only (ours) |
|---|---|---|---|---|---|
| AUROC(%) for classification. Higher is better. Majority baseline is 50.0. | | | | | |
| rel-amazon | item-churn | 67.6 | **72.5** | +4.9 | 71.0 |
| rel-amazon | user-churn | 64.2 | **65.0** | +0.8 | 64.4 |
| rel-avito | user-clicks | 54.7 | 47.9 | −6.8 | 45.9 |
| rel-avito | user-visits | 57.2 | **63.4** | +6.2 | 63.5 |
| rel-f1 | driver-dnf | 80.7 | **81.0** | +0.3 | 76.7 |
| rel-f1 | driver-top3 | 86.9 | **88.4** | +1.5 | 82.6 |
| rel-hm | user-churn | **67.4** | 66.0 | −1.4 | 63.7 |
| rel-stack | user-badge | 80.0 | **82.0** | +2.0 | 81.4 |
| rel-stack | user-engage | 78.9 | **86.2** | +7.4 | 82.4 |
| rel-trial | study-out | **54.4** | 51.8 | −2.6 | 53.8 |
| Mean | | 69.2 | **70.4** | +1.2 | 68.5 |
| $R^2$(%) for regression. Higher is better. Mean baseline is 0.0. | | | | | |
| rel-amazon | item-ltv | 35.3 | **40.5** | +5.2 | 10.7 |
| rel-amazon | user-ltv | 14.5 | **18.5** | +4.0 | 9.8 |
| rel-avito | ad-ctr | 3.1 | **4.9** | +1.9 | 2.5 |
| rel-f1 | driver-pos | 54.3 | **55.5** | +1.2 | 41.3 |
| rel-hm | item-sales | 16.0 | **20.0** | +4.0 | 4.4 |
| rel-stack | post-votes | 22.3 | **25.5** | +3.2 | 15.7 |
| rel-trial | site-succ | 33.7 | **38.6** | +5.0 | 38.3 |
| rel-trial | study-adv | **1.9** | 1.6 | −0.3 | −0.8 |
| Mean | | 22.6 | **25.7** | +3.0 | 15.2 |

*Table 1.* Zero-shot test set results on unseen datasets for different pretraining setups. **Real only** pretraining is done with RelBench in a leave-one-DB-out setting. **Synthetic only** pretraining is done on PLUREL generated synthetic data. **Synthetic + Real** involves continued pretraining on RelBench (leave-one-DB-out) from the checkpoint obtained with **Synthetic only** pretraining. First 2 columns report mean over 3 seeds. See Appendix D.1 (Table 4) for standard error and results for a different base model.

bined with real-data continued pretraining. On average, **Synthetic+Real** achieves a $+1.2\%$ absolute gain in AUROC and a $+3.0\%$ absolute gain in $R^2$ over the **Real only** baseline, reaching up to $+7.4\%$ and $+5.2\%$ respectively on individual tasks. Improvements are particularly strong on regression tasks, where **Synthetic+Real** outperforms **Real only** on 7 out of 8 tasks, indicating that synthetic relational diversity is especially beneficial for learning continuous-valued patterns. For classification tasks, gains are more mixed but remain positive on average, with large improvements observed on behavior-driven tasks such as user-engage and item-ltv. In contrast, **Synthetic only** underperforms both baselines on most tasks, highlighting that synthetic data alone is insufficient for robust zero-shot transfer and that continued pretraining on real data is critical for distribution alignment. On certain tasks we observe a slight decrease in zero-shot performance when starting with synthetic data. We hypothesize that this is due to the lack of textual information and column semantics in PLUREL.

## 4. Related Work

**Foundation Models.** In recent years, the machine learning community has achieved significant advances through the development of foundation models trained on massive, diverse datasets (Bommasani et al., 2022). These models serve as versatile backbones for continued training and can be di-

rectly applied to new problems in few-shot settings (Zhou et al., 2024). While vast amounts of publicly crawled text and image data have enabled the continued advancement of frontier language and vision models (Achiam et al., 2023; Team et al., 2025; Yang et al., 2025), a sharp contrast exists in the relational domain. Relational databases are rarely public, as they typically contain sensitive user or enterprise information (Patki et al., 2016). Consequently, concerns over data privacy and the lack of truly massive public and diverse datasets suitable for pretraining have hindered the development of RFMs. We approach this issue by proposing a framework capable of generating diverse pretraining data, free of PII or confidentiality, from scratch.

**Synthetic Data and Tabular Foundation Models.** Synthetic data offers a promising alternative. Hollmann et al. (2023) introduces TabPFN, a transformer for in-context learning on tabular data pretrained on millions of synthetic tabular datasets. The method proposes a synthetic data-generating process based on SCMs (Müller et al., 2022) that is capable of capturing causal relationships between columns observed in real-world tabular data. Later works combine SCMs with tree-based data generators using decision tree (den Breejen et al., 2025) and XGBoost-based (QU et al., 2025) generators. Zhang et al. (2025) identify two key properties of these generators that enable strong generalization in pretrained TFMs: (i) the scale of the pretraining data, and (ii) the diversity of the generated datasets. However, the essence of relational data lies in inter-table primary–foreign key relationships (Fey et al., 2024; Dwivedi et al., 2025), therefore our work extends and generalizes previous efforts to multi-tabular settings.

**Relational Foundation Models.** For relational learning, no prior work has proposed a synthetic generator designed to facilitate pretraining of RFMs. Recent works such as Griffin (Wang et al., 2025), and the Relational Transformer (RT) (Ranjan et al., 2026) develop RFMs pretrained on real-world data. Griffin relies in large part on single-table pretraining and utilizes only 14 databases from the 4DBInfer (Wang et al., 2024) and RelBench (Robinson et al., 2024) collection. Whereas RT utilizes only 6 databases from RelBench, resulting in a limited pretraining corpus. Alternatively some works repurpose TFMs for graph settings (Eremeev et al., 2025; Hayler et al., 2025), and benefit from further training on multi-table or graph datasets. The enterprise model KumoRFM (Fey et al., 2025) utilizes a mix of publicly available databases and synthetic data for pretraining, but the details of the datasets remain undisclosed. Our work addresses these shortcomings by providing an accessible framework to generate diverse pretraining data.

**Scaling Laws.** The development of ever larger foundation models is driven by the promise of scaling laws, which predict improvements in model performance as a function of increasing data and model sizes (Kaplan et al., 2020). In the language and vision domains, established scaling laws characterize performance as a function of dataset size, model size, and compute (Hoffmann et al., 2022; Zhai et al., 2022). Schambach et al. (2023) analyzes the scaling behavior of tabular models, while Ma et al. (2025) provides explicit scaling laws characterizing the training of TFMs with respect to model size and the number of cells in the training corpus. Zhang et al. (2025) examines the scaling behaviour of TFMs trained on synthetic data and identifies diversity of the generated data as a key property enabling generalization. However, no prior work has examined the scaling of RFMs. PLUREL addresses this gap and allows us to provide RFM scaling laws not only for dataset size but also for data diversity, quantified by the number of synthetic databases in pretraining.

**Relational Database Generation.** Another related line of research is privacy-preserving synthetic database generation (Patki et al., 2016). These methods focus on reproducing the structure and statistical properties of a given real-world database while protecting the privacy of the data subjects. Similar statistical approaches have also been considered to synthesize relational databases for testing and benchmarking the performance of analytical applications (Houkjær et al., 2006; Arasu et al., 2011). Recent works propose approaches based on graphical models (Cai et al., 2023), generative adversarial networks (Gueye et al., 2023), transformers (Solatorio & Dupriez, 2023), diffusion (Pang et al., 2024; Hudovernik, 2024) and graph-based models (Scassola et al., 2025; Ketata et al., 2025; Hudovernik et al., 2025). While these approaches address privacy concerns and can facilitate broader data sharing, they remain tied to existing real-world databases. They require real databases as input and are constrained to generating new samples that conform to the original schema. Further they are computationally expensive for large-scale data generation. In contrast, PLUREL provides a lightweight framework for generating diverse schemas, row-connectivity patterns, and feature distributions, unlocking data scaling for RFMs.

## 5. Conclusion and Future Work

In this work, we introduce PLUREL, a novel framework for generating synthetic relational databases from scratch for Relational Foundation Model (RFM) pretraining. PLUREL offers a flexible design space capable of synthesizing diverse databases, and unlocks large-scale synthetic pretraining without privacy constraints. Through experiments with the Relational Transformer (RT), we find that (1) pretraining loss exhibits a power-law trend as the number of synthetic databases and dataset size increase, (2) models pretrained on larger and more diverse synthetic datasets generalize more effectively to previously unseen real data,

and (3) synthetic pretraining produces robust base models that enhance subsequent pretraining on real data.

Our framework and results open several new directions of research: (1) relational data curation and synthetic design space exploration, (2) extending PLUREL to additional data types such as text, (3) semi-synthetic data augmentation to expand real-world databases, (4) pretraining curriculums and strategies to combine synthetic and real data, (5) exploring impact of synthetic data on long-context modeling and test-time scaling, and (6) joint model- and data-scaling laws. By unlocking scalable pretraining data for RFMs, PLUREL sets the stage for their broader applicability across domains.

## Impact Statement

This paper presents PLUREL, a new framework for generating synthetic relational databases from scratch, aimed at addressing the scarcity of diverse, public available relational data for training Relational Foundation Models (RFMs). By enabling the synthesis of unlimited relational databases with configurable schemas, connectivity patterns, and data distributions, our work contributes to the broader field of Foundation Models in AI, and relational deep learning, offering a privacy-preserving approach to developing AI systems on real-world enterprise data. We do so while unlocking new scaling laws for this field, analogous to those observed in language, vision and other data domains.

The societal impact of this work aligns with the broader advancements in foundation models and enterprise AI, with potential applications in business intelligence, fraud detection, consumer analytics, healthcare, and supply chain industries. Our work has profound implications for maintaining the privacy of global consumer data, as no real business or consumer data is required for foundation model development when the proposed PLUREL is used. By democratizing access to large-scale relational pretraining data, PLUREL could accelerate the development of RFMs that benefit organizations of all sizes, especially reducing barriers to AI adoption for enterprises that lack extensive proprietary databases.

## Acknowledgments

We thank Tom Palczewski, Charilaos Kanatsoulis, Nils Walter, Shirley Wu, Jonas De Schouwer, Harshvardhan Agarwal, Marcel Roed, Michael Bereket, Rok Sosic, Yanay Rosen, Moritz Schaefer, Tianlang Chen, Anvita Gupta, Mark Li, Sam Thelin, Mahmoud Mohammadi, Joe Meyer, and Roshan Reddy for discussions and for providing feedback on our manuscript. We also gratefully acknowledge the support of NSF under Nos. CCF-1918940 (Expeditions), DMS-2327709 (IHBEM), IIS-2403318 (III); NIH under No. 1U24NS146314-01, Stanford Data Applications Initiative, Wu Tsai Neurosciences Institute, Stanford Institute for Human-Centered AI, Chan Zuckerberg Initiative, Amazon, Genentech, SAP, and SCBX. The content is solely the responsibility of the authors and does not necessarily represent the official views of the funding entities.

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

## A. Limitations

Currently, PLUREL supports primary–foreign key (P→F) connectivity between rows of a table $T$ and a (different) parent table $\widetilde{T} \neq T$ in the schema graph $\mathcal{G}$. However, certain real-world databases may exhibit self-loops in their P→F connectivity. For example, the `ParentID` column (F) in the `posts` table of `rel-stack` [2] refers to the `Id` column (P) of the same table. Modeling such self-loops through SCMs is currently not supported and is an interesting extension of the framework.

## B. Synthesizing Databases with PLUREL

In Section 2, we introduced PLUREL in its most generic form. In this section, we detail the design choices and hyperparameter distributions, along with the algorithms describing each stage. A summary is presented in Table 2.

| | Parameter | Kind | Sampling | Choices |
|---|---|---|---|---|
| **Database** | Schema graph priors ($\mathcal{P}_G$) | set | uniform | {Barabasi-Albert, Reverse Random-Tree, Watts-Strogatz} |
| | Num tables | range | uniform | [3, 20] |
| | Num rows (*entity tables*) | range | uniform | [500, 1000] |
| | Num rows (*activity tables*) | range | uniform | [2000, 5000] |
| | Num columns | range | power-law | [3, 40] |
| | Min timestamp | constant | - | 1990-01-01 |
| | Max timestamp | constant | - | 2025-01-01 |
| | NULL cells (%) | range | uniform | [0.01, 0.1] |
| **Table / SCM** | SCM causal graph prior ($\mathcal{P}_C$) | set | uniform | {Layered, Erdos-Renyi, Barabasi-Albert, Random-Tree, Reverse Random-Tree} |
| | SCM feature node % | range | uniform | [0.3, 0.9] |
| | Num categories | range | uniform | [2, 10] |
| | MLP initializations | set | uniform | { kaiming normal, kaiming uniform, xavier normal, xavier uniform, trunc normal, sparse(0.5) } |
| | MLP activations | set | uniform | { relu, elu, silu, softsign, tanh } |
| | MLP input dimension | constant | - | 1 |
| | MLP hidden dimension | constant | - | 32 |
| | MLP output dimension | constant | - | 1 |
| | MLP depth | constant | - | 2 |
| | Exogenous input prior ($\xi$) | set | uniform | { Beta(0.5, 0.5), Beta(2.0, 2.0), Beta(2.0, 3.0), Beta(2.0, 4.0), Beta(4.0, 1.0) } |
| | HSBM levels | range | uniform | [1, 5] |
| | HSBM clusters per level | range | uniform | [1, 3] |
| | Temporal trend exponent | range | uniform | [0, 2] |
| | Temporal trend scale (*activity table*) | set | uniform | [−1, 1] |
| | Temporal trend scale (*entity table*) | constant | - | 0.0 |
| | Temporal cycle frequency | set | uniform | {0.1, 0.2, 0.3, 0.4, 0.5, 0.6, 0.7, 0.8, 0.9, 1.0} |
| | Temporal cycle scale (*activity table*) | set | uniform | [−1, 1] |
| | Temporal cycle scale (*entity table*) | constant | - | 0.0 |
| | Temporal noise scale (*activity table*) | constant | - | 0.05 |
| | Temporal noise scale (*entity table*) | constant | - | 1.0 |
| **DAG** | Barabasi–Albert: edge dropout | constant | - | 0.4 |
| | Barabasi–Albert: node attachment edges | constant | - | 2 |
| | Erdos–Renyi: edge probability | range | uniform | [0.3, 0.8] |
| | Watts-Strogatz: rewire probability | constant | uniform | [0.1, 0.3] |
| | Layered: number of levels (depth) | range | uniform | [2, 8] |
| | Layered: edge dropout | constant | - | 0.1 |

*Table 2.* Design choices and the distribution of PLUREL hyperparameters.

### B.1. Stage 1: Schema Generation via Directed Graphs

The schema graph $\mathcal{G}$ can be sampled from any class of directed graphs with an arbitrary number of nodes (representing tables). However, the role of $\mathcal{G}$ extends beyond this layer of abstraction. It determines the fraction of tokens being used from the same table, row, column, parent tables, and child tables for preparing the context of the foundation model being developed, which, in this work, is the Relational Transformer (RT) (Ranjan et al., 2026). This context informs RT about the relational attention patterns to learn, which in turn affects its zero-shot performance on unseen databases. To this end, we choose the `Barabasi-Albert` (BA), `Reverse Random-Tree` (RRT), and the `Watts-Strogatz` (WS) family of DAGs as the graph priors. BA graphs model RDBs with hub tables and preferential connectivity between tables. RRT graphs model a hierarchy of tables, and WA graphs model RDBs with table clusters. We sparsify and rewire edges for BA and WS graphs, respectively, to increase diversity. The pseudocode is presented in Algorithm 1.

**Metadata.** After sampling $\mathcal{G}$, we assign each table $T$ its type (*entity* or *activity*), the number of columns $F_T$, and rows $R_T$. Using this metadata, the primary key column is named as `row_idx`, the feature columns are named as `feature_i`, where $i \in \{1, \cdots, F_T\}$, and the foreign key columns are named as `foreign_row_t`, with $t \in \{1, \cdots, |\Pr(T, \mathcal{G})|\}$.

---

[2]https://relbench.stanford.edu/datasets/rel-stack/

---

**Algorithm 1** Schema generation

---

**Input:** number of tables $|\mathcal{T}|$, graph prior $\mathcal{P}_\mathcal{G}$
**Output:** schema graph $\mathcal{G}$, table metadata

1: Sample a directed graph $\mathcal{G} \sim \mathcal{P}_\mathcal{G}$ over nodes $\mathcal{T}$
2: Apply sparsification/rewiring on $\mathcal{G}$
3: **for** each table $T$ in the topological ordering of nodes $\mathcal{T}$ **do**
4:     Set foreign key columns according to $|\Pr(T, \mathcal{G})|$
5:     Sample number of feature columns
6:     **if** out_deg$(T, \mathcal{G}) \geq 1$ **then**
7:         Assign $T$ as an *entity table*
8:     **else**
9:         Assign $T$ as an *activity table*
10:     **end if**
11:     Sample number of rows conditioned on table type
12: **end for**
13: **return** $\mathcal{G}$ and table metadata

---

## B.2. Stage 2: Foreign Key Generation via Bipartite Graphs

In this stage, we first populate the primary key values of $T$ as its row indices. Considering a parent table $\widetilde{T} \in \mathtt{Pr}(T, \mathcal{G})$, we cluster the rows of $T, \widetilde{T}$ into a hierarchy of blocks $\mathbf{H}_T = (B_T^1, \cdots B_T^L)$ and $\mathbf{H}_{\widetilde{T}} = (B_{\widetilde{T}}^1, \cdots B_{\widetilde{T}}^L)$ respectively. The number of HSBM levels $L$ is chosen uniformly from $[1, 5]$ with the size of each block $B_T^l, B_{\widetilde{T}}^l$ chosen uniformly from $[1, 3]$. We sample the entries of the block connectivity matrix (Equation (1)) $\mathbf{P}[l], \forall l \in [L]$ as follows:

$$\mathbf{P}[l]_{ij} = \begin{cases} 0.9, & \text{if } i \equiv j \pmod{\min(B_{\widetilde{T}}^l, B_T^l)}, \\ U(0.001, \ 0.002), & \text{otherwise.} \end{cases} \tag{9}$$

This ensures that rows of $T$ and $\widetilde{T}$ from the same level and block index (modulo) are preferentially connected and form well-separated clusters. For each row of table $T$, we use Equation 2 to sample the primary key $j$ of table $\widetilde{T}$ and assign it to the corresponding foreign key column in $T$. The pseudocode is presented in Algorithm 2.

---

**Algorithm 2** Foreign key generation

---

**Input:** table $T$, parent table $\widetilde{T}$, number of rows $R_T, R_{\widetilde{T}}$
**Output:** foreign key column $\text{fk}_{T \leftarrow \widetilde{T}}$

1: Set primary keys of $T$ and $\widetilde{T}$ as row indices $\{1, \ldots, R_T\}$ and $\{1, \ldots, R_{\widetilde{T}}\}$
2: Cluster rows of $T$ into hierarchy of blocks $\mathbf{H}_T$
3: Cluster rows of $\widetilde{T}$ into hierarchy of blocks $\mathbf{H}_{\widetilde{T}}$
4: Sample HSBM probability matrix $\mathbf{P}$ over $\mathbf{H}_T, \mathbf{H}_{\widetilde{T}}$
5: **for** each row $i \in T$ **do**
6:     Sample parent row index $j \in \widetilde{T}$ according to the HSBM-induced block connectivity
7:     Set $\text{fk}_{T \leftarrow \widetilde{T}}[i] \leftarrow j$
8: **end for**
9: **return** foreign key column $\text{fk}_{T \leftarrow \widetilde{T}}$

---

## B.3. Stage 3: Feature Generation via Structural Causal Models

As described in Section 2.3, each table $T \in \mathcal{T}$ is associated with an SCM. We sample the causal graph $\mathcal{C}$ from the $\{$`Layered`, `Erdos-Renyi`, `Barabasi-Albert`, `Random-Tree`, `Reverse Random-Tree`$\}$ families to model diverse causal relationships between latent and feature nodes. The exogenous input of source nodes for activity tables is modeled with trend and cyclical patterns, along with random normal fluctuations. Whereas the exogenous input of source nodes for entity tables is only modeled with random normal fluctuations. The intuition is that entity tables model static users or items and

therefore do not necessarily exhibit temporal correlations among features. Nonetheless, it is an experimental design, and not a limitation of the framework itself. The exogenous input for non-source nodes (as used in Equation (5)) is an $\mathbb{R}^{d_{\text{hid}}}$ vector with $d_{\text{hid}} = 32$ and each entry sampled from a $\texttt{Beta}$ distribution chosen from $\{$ $\texttt{Beta(0.5, 0.5)}, \texttt{Beta(2.0,}$ $\texttt{2.0)}, \texttt{Beta(2.0, 3.0)}, \texttt{Beta(2.0, 4.0)}, \texttt{Beta(4.0, 1.0)}$ $\}$. In the causal graph $\mathcal{C}$, we assign edge weights by sampling from a normal distribution $\mathcal{N}(0, 1)$. These weights are used to aggregate the embeddings of predecessor nodes $\texttt{Pr}(v_i, \mathcal{C}_T)$ in Equation (5) during value propagation through $\mathcal{C}_T$. For aggregating embeddings of feature nodes originating from foreign SCMs, we instead use a uniform weight of $1/|\mathcal{V}_{\tilde{T}}^F|$. The pseudocode is presented in Algorithm 3.

---

**Algorithm 3** Feature generation

---

**Input:** table $T$, parent tables $\texttt{Pr}(T, \mathcal{G})$, row count $R_T$, SCM $(\mathcal{C}_T, \mathcal{Z}_T)$ with feature nodes $\mathcal{V}_T^F$, source nodes $\mathcal{V}_T^S$
**Output:** populated table $T$

1: **for** each row index $r = 1$ to $R_T$ **do**
2:     Initialize exogenous inputs $\mathbf{u}_i^{(r)}$ for all source nodes $v_i \in \mathcal{V}_T^S$ using temporal patterns
3:     Assign values to source nodes using $z_i = H_i(\mathbf{u}_i^{(r)})$
4:     **for** each non-source node $v_i$ in topological order of $\mathcal{C}_T$ **do**
5:         Collect predecessor node values $\texttt{Pr}(v_i, \mathcal{C}_T)$
6:         Collect feature values from parent-table SCMs indexed by foreign keys
7:         Project collected values into a shared latent space
8:         Aggregate projected representations with exogenous input according to the SCM mechanism
9:         Reconstruct a type-specific value for $v_i$
10:     **end for**
11:     Write values of feature nodes $\mathcal{V}_T^F$ to the cells of row $r$ in table $T$
12: **end for**
13: **return** populated table $T$

---

### B.4. Computational Efficiency

To characterize the computational footprint of PLUREL, we generate synthetic RDBs with varying numbers of tables using a single-threaded process. For each configuration, other hyperparameters are fixed as in Table 2 and results are averaged over ten random seeds. As shown in Table 3, generation latency increases approximately linearly with the number of tables, ranging from 2.5 seconds for 10 tables to 26.1 seconds for 80 tables on average, corresponding to an average throughput of roughly 0.2–0.4 seconds per table. Peak memory usage is around 1 GB even in the largest setting of 80 tables/RDB. The dominant contributor to end-to-end latency is conditional row generation induced by primary–foreign key connectivity, while schema graph instantiation and post-processing incur comparatively minor overhead. The sparsity of the schema graph $\mathcal{G}$ plays a central role in determining latency. Since the pipeline is CPU-only, PLUREL introduces minimal overhead relative to downstream GPU training and remains suitable for both low-resource environments and datacenter-scale deployments.

| Number of Tables | Latency (sec) | Peak Memory (GB) |
|:---:|:---:|:---:|
| 10 | $2.5_{\pm 0.8}$ | $0.76_{\pm 0.23}$ |
| 20 | $5.6_{\pm 2.4}$ | $0.94_{\pm 0.26}$ |
| 40 | $13.0_{\pm 8.3}$ | $1.14_{\pm 0.35}$ |
| 80 | $26.1_{\pm 12.6}$ | $1.35_{\pm 0.40}$ |

*Table 3.* Latency (in seconds) and peak memory (GB) required to generate a varying number of tables in a single synthetic RDB with a single-threaded process. The mean and standard deviation are computed across 10 seeds. The large variance in latency is a result of diverse schema graphs being sampled across the seeds, with sparser graphs resulting in faster table generation.

## C. Background on Relational Transformer

The Relational Transformer (RT) (Ranjan et al., 2026) is a specialized transformer architecture for modeling relational data and enabling zero-shot generalization to predictive tasks on unseen databases. It achieves this with two key designs: (1) *cell-level tokenization* of relational data, and (2) a *Relational Attention* mechanism. RT outperforms state-of-the-art LLMs on predictive tasks with its zero-shot generalization capabilities and introduces a new modeling paradigm for RFMs.

### C.1. Token Representations

RT represents a relational database as a sequence of cells, with each cell $(v, c, t)$ represented by a single token. Here $v$ is the cell value, $c$ is the column name, and $t$ is the table name. RT employs type-specific processing to normalize numeric, boolean, and datetime type cells and project these modalities into a shared embedding space. Text-type cells are first embedded using a frozen text encoder and projected into this shared space. By integrating the predictive task as a designated table in the database, RT unifies all downstream tasks to be cast as a Masked Token Prediction (MTP) objective, supporting scalable self-supervised learning. Finally, to incorporate schema semantics, RT embeds column and table names by embedding the phrase "<column name> of <table name>" (e.g., "price of product") using a pretrained sentence encoder. The final token embedding is obtained by projecting these normalized values via a data-type-specific weight matrix and adding the projected embeddings. For cells masked during MTP, the value embedding is replaced by a learned, data-type-specific mask vector.

### C.2. Relational Attention

RT operates on cell-level tokens to model dependencies across rows, columns, and tables. The architecture augments standard transformer blocks with *Relational Attention*, comprising three structured attention layers followed by a bidirectional attention layer. It is implemented using the *masked scaled dot-product attention (SDPA)* as

$$\text{Attention}(\mathbf{Q}, \mathbf{K}, \mathbf{V}; \mathbf{M}) = \text{Softmax}\left(\frac{\text{Mask}(\mathbf{Q}\mathbf{K}^\top; \mathbf{M})}{\sqrt{d_k}}\right)\mathbf{V}, \qquad \text{Mask}(\mathbf{A}; \mathbf{M})_{ij} = \begin{cases} \mathbf{A}_{ij}, & \mathbf{M}_{ij} = 1 \\ -\infty, & \mathbf{M}_{ij} = 0. \end{cases} \tag{10}$$

Here $\mathbf{Q}, \mathbf{K} \in \mathbb{R}^{n \times d_k}$ and $\mathbf{V} \in \mathbb{R}^{n \times d_v}$ denote the query, key, and value matrices, where $n$ is the context length. The binary mask $\mathbf{M} \in \{0, 1\}^{n \times n}$ specifies allowable token interactions, with $\mathbf{M}[q, k] = 1$ indicating that token $q$ can attend to token $k$. For example, causal language models use $\mathbf{M}^{\text{causal}}[q, k] = \mathbf{1}\{k \leq q\}$. **Column Attention.** Restricts attention to tokens within the same column, capturing attribute-level statistics and cross-row patterns. **Feature Attention.** Allows attention within the same row and to parent rows connected via foreign–primary key (F→P) links, aggregating attributes that describe a given entity. **Neighbor Attention.** Enables attention to child rows linked via primary–foreign (P→F) keys, analogous to message passing in graph neural networks. Finally, **Full attention** is a standard bidirectional layer allowing pairwise interactions between all the tokens. However, due to its limited utility on RelBench tasks as observed by (Ranjan et al., 2026), we skip this layer in our RT models. Furthermore, owing to the diverse modalities of data, RT with the standard Relational Attention layer exhibits early overfitting behaviour during pretraining. We address this key gap in Appendix D.2.

### C.3. Context Preparation with Breadth First Search (BFS) Sampling

For a given seed row, which is typically a row in the task table, RT independently constructs a context window anchored at this seed row and expands it to a fixed budget of $L$ cells using a relation-aware, bounded breadth-first traversal. Rows serve as the sampling unit, where once a row is selected, all feature cells (other than primary/foreign key columns) are added to the context. Starting from the seed row, the algorithm traverses foreign–primary (F→P) and primary–foreign (P→F) key links, prioritizing low-hop neighbors under the assumption that proximity in the relational graph correlates with relevant information for predictions. To control graph expansion, F→P links are always followed immediately, whereas P→F links are subsampled by enforcing a maximum fan-out of $w$ child rows per parent. The traversal terminates when the total number of collected cells reaches the context budget. Furthermore, rows with timestamps greater than that of the seed row are excluded from the context to enforce temporal consistency. We refer to Ranjan et al. (2026) for additional details.

## D. Additional Experiments

### D.1. Error Bars for Main Experiments

In Table 4 we report uncertainty estimates from our zero-shot evaluation of different pretraining strategies reported in Table 1. We report the mean and standard error across three random seeds. When pretrained using synthetic data followed by real data, the model consistently improves upon the performance of RT trained solely on real data. On certain tasks, we observe slight degradations in performance. Notably, these tasks align with those for which the original authors report performance degradation when ablating table semantics (see Ranjan et al. (2026), Appendix E, Table 8). Since row values generated by PLUREL do not functionally depend on table semantics (i.e., column and table names), we hypothesize that this limitation contributes to the observed performance drop.

**Synthetic Pretraining: 1024 RDBs, 4B tokens.**

| Dataset | Task | Real only | Synthetic + Real (ours) | Absolute Gain (%) | Synthetic only (ours) |
|---|---|---|---|---|---|
| AUROC(%) for classification. Higher is better. Majority baseline is 50.0. | | | | | |
| rel-amazon | item-churn | $67.6_{\pm0.8}$ | $\mathbf{72.5}_{\pm0.1}$ | +4.9 | 71.0 |
| rel-amazon | user-churn | $64.2_{\pm0.1}$ | $\mathbf{65.0}_{\pm0.0}$ | +0.8 | 64.4 |
| rel-avito | user-clicks | $\mathbf{54.7}_{\pm2.9}$ | $47.9_{\pm1.0}$ | −6.8 | 45.9 |
| rel-avito | user-visits | $57.2_{\pm2.8}$ | $\mathbf{63.4}_{\pm0.0}$ | +6.2 | 63.5 |
| rel-f1 | driver-dnf | $80.7_{\pm0.4}$ | $\mathbf{81.0}_{\pm0.5}$ | +0.3 | 76.7 |
| rel-f1 | driver-top3 | $86.9_{\pm0.4}$ | $\mathbf{88.4}_{\pm0.0}$ | +1.5 | 82.6 |
| rel-hm | user-churn | $\mathbf{67.4}_{\pm0.2}$ | $66.0_{\pm0.2}$ | −1.4 | 63.7 |
| rel-stack | user-badge | $80.0_{\pm1.1}$ | $\mathbf{82.0}_{\pm0.3}$ | +2.0 | 81.4 |
| rel-stack | user-engage | $78.9_{\pm1.4}$ | $\mathbf{86.2}_{\pm0.0}$ | +7.4 | 82.4 |
| rel-trial | study-out | $\mathbf{54.4}_{\pm1.2}$ | $51.8_{\pm2.6}$ | −2.6 | 53.8 |
| Mean | | $69.2_{\pm0.6}$ | $\mathbf{70.4}_{\pm0.3}$ | +1.2 | 68.5 |
| $R^2$(%) for regression. Higher is better. Mean baseline is 0.0. | | | | | |
| rel-amazon | item-ltv | $35.3_{\pm3.3}$ | $\mathbf{40.5}_{\pm0.6}$ | +5.2 | 10.7 |
| rel-amazon | user-ltv | $14.5_{\pm1.2}$ | $\mathbf{18.5}_{\pm1.7}$ | +4.0 | 9.8 |
| rel-avito | ad-ctr | $3.1_{\pm0.3}$ | $\mathbf{4.9}_{\pm1.3}$ | +1.9 | 2.5 |
| rel-f1 | driver-pos | $54.3_{\pm0.6}$ | $\mathbf{55.5}_{\pm0.5}$ | +1.2 | 41.3 |
| rel-hm | item-sales | $16.0_{\pm0.8}$ | $\mathbf{20.0}_{\pm1.4}$ | +4.0 | 4.4 |
| rel-stack | post-votes | $22.3_{\pm2.2}$ | $\mathbf{25.5}_{\pm0.1}$ | +3.2 | 15.7 |
| rel-trial | site-succ | $33.7_{\pm0.5}$ | $\mathbf{38.6}_{\pm0.2}$ | +5.0 | 38.3 |
| rel-trial | study-adv | $\mathbf{1.9}_{\pm0.8}$ | $1.6_{\pm0.2}$ | −0.3 | −0.8 |
| Mean | | $22.6_{\pm0.6}$ | $\mathbf{25.7}_{\pm0.1}$ | +3.0 | 15.2 |

**Synthetic Pretraining: 512 RDBs, 16B tokens.**

| Dataset | Task | Real only | Synthetic + Real (ours) | Absolute Gain (%) | Synthetic only (ours) |
|---|---|---|---|---|---|
| AUROC(%) for classification. Higher is better. Majority baseline is 50.0. | | | | | |
| rel-amazon | item-churn | $67.6_{\pm0.8}$ | $\mathbf{72.5}_{\pm0.4}$ | +4.8 | 69.0 |
| rel-amazon | user-churn | $64.2_{\pm0.1}$ | $\mathbf{64.7}_{\pm0.1}$ | +0.5 | 64.1 |
| rel-avito | user-clicks | $\mathbf{54.7}_{\pm2.9}$ | $50.0_{\pm0.9}$ | −4.7 | 46.4 |
| rel-avito | user-visits | $57.2_{\pm2.8}$ | $\mathbf{62.2}_{\pm0.1}$ | +5.0 | 62.3 |
| rel-f1 | driver-dnf | $80.7_{\pm0.4}$ | $\mathbf{81.4}_{\pm0.2}$ | +0.8 | 77.6 |
| rel-f1 | driver-top3 | $86.9_{\pm0.4}$ | $\mathbf{88.6}_{\pm0.2}$ | +1.7 | 81.4 |
| rel-hm | user-churn | $\mathbf{67.4}_{\pm0.2}$ | $66.5_{\pm0.7}$ | −0.9 | 63.1 |
| rel-stack | user-badge | $80.0_{\pm1.1}$ | $\mathbf{82.0}_{\pm0.2}$ | +2.0 | 77.0 |
| rel-stack | user-engage | $78.9_{\pm1.4}$ | $\mathbf{85.2}_{\pm0.2}$ | +6.4 | 71.5 |
| rel-trial | study-out | $\mathbf{54.4}_{\pm1.2}$ | $51.6_{\pm0.4}$ | −2.9 | 55.1 |
| Mean | | $69.2_{\pm0.6}$ | $\mathbf{70.5}_{\pm0.0}$ | +1.3 | 66.8 |
| $R^2$(%) for regression. Higher is better. Mean baseline is 0.0. | | | | | |
| rel-amazon | item-ltv | $35.3_{\pm3.3}$ | $\mathbf{39.4}_{\pm0.5}$ | +4.1 | 9.7 |
| rel-amazon | user-ltv | $14.5_{\pm1.2}$ | $\mathbf{21.7}_{\pm0.7}$ | +7.2 | 9.2 |
| rel-avito | ad-ctr | $3.1_{\pm0.3}$ | $\mathbf{7.9}_{\pm0.9}$ | +4.9 | 2.0 |
| rel-f1 | driver-pos | $54.3_{\pm0.6}$ | $\mathbf{54.7}_{\pm0.3}$ | +0.5 | 35.9 |
| rel-hm | item-sales | $16.0_{\pm0.8}$ | $\mathbf{25.6}_{\pm1.0}$ | +9.5 | 5.4 |
| rel-stack | post-votes | $22.3_{\pm2.2}$ | $\mathbf{24.7}_{\pm0.6}$ | +2.4 | 14.1 |
| rel-trial | site-succ | $33.7_{\pm0.5}$ | $\mathbf{37.9}_{\pm0.2}$ | +4.2 | 35.3 |
| rel-trial | study-adv | $\mathbf{1.9}_{\pm0.8}$ | $1.3_{\pm0.5}$ | −0.6 | −0.7 |
| Mean | | $22.6_{\pm0.6}$ | $\mathbf{26.7}_{\pm0.2}$ | +4.0 | 13.9 |

*Table 4.* Same setup as Table 1. Here we also report $\pm$ standard error over 3 seeds. Continued pretraining is robust to choice of base model. Worse synthetic-only column can still give better results in the synthetic+real column (*e.g.,* compare the Mean rows between the two tables) indicating the occurrence of post-hoc reversal (Ranjan et al., 2024) and suggesting that post-hoc model selection would be ideal.

### D.2. Architectural Improvements: Query-Key Normalization

The RT architecture supports multi-modal input representations for text, numeric, and boolean cell tokens, with type-specific encoders. During synthetic pretraining with such multi-modal(type) input tokens, we observed that zero-shot generalization to RelBench tasks was sensitive to the RT initialization. To reduce such sensitivity, we applied Query-Key Normalization (QK-Norm) (Henry et al., 2020; Wortsman et al., 2024; Team, 2025) with RMSNorm across the head dimension (per head) to the relational attention layer (10). Formally, the masked scaled dot-product attention with QK-Norm is given by:

$$\text{Attention}(\mathbf{Q}, \mathbf{K}, \mathbf{V}; \mathbf{M}) = \text{Softmax}\left(\frac{\text{Mask}(\text{RMSNorm}\,(\mathbf{Q})\,\text{RMSNorm}\,(\mathbf{K})^{\top}; \mathbf{M})}{\sqrt{d_k}}\right)\mathbf{V} \qquad (11)$$

**Reducing variance across seeds.** We initialized RT with four different seeds $\{0, 1, 2, 3\}$ and used the same seed $(0)$ for the training and evaluation data loaders. We pretrain RT in `BFloat16` precision with synthetic data on 1B tokens with the rest of the hyperparameters chosen as per Section 3. We use the `rel-amazon` tasks in RelBench for measuring zero-shot generalization. Without QK Norm, the maximum AUROC $(\%)$ difference across model seeds on the test split of `rel-amazon/item-churn` task was as high as $9.4\%$ at the end of training. Furthermore, the difference was even higher $(10.5\%)$ on the test split of the `rel-amazon/user-churn` task. With QK Norm, such sensitivity to initialization is mitigated, and the difference across seeds reduces to $3.4\%$ and $2.2\%$ respectively.

**Effects on baseline performance.** Following the same setup as Section 3.3, we pretrained a randomly initialized RT without QK-Norm on RelBench data using the *leave-one-db-out* approach and noticed a drop in the baseline performance. In particular, without QK-Norm, RT suffers from an early overfitting problem on certain tasks (especially binary classification), while also lowering the peak performance (see Figure 5). We also observed that the baseline (Real only) mean test AUROC and R $^2$ (%) can decrease by $3.1\%$ (absolute) and $3.7\%$ (absolute) without QK Norm.

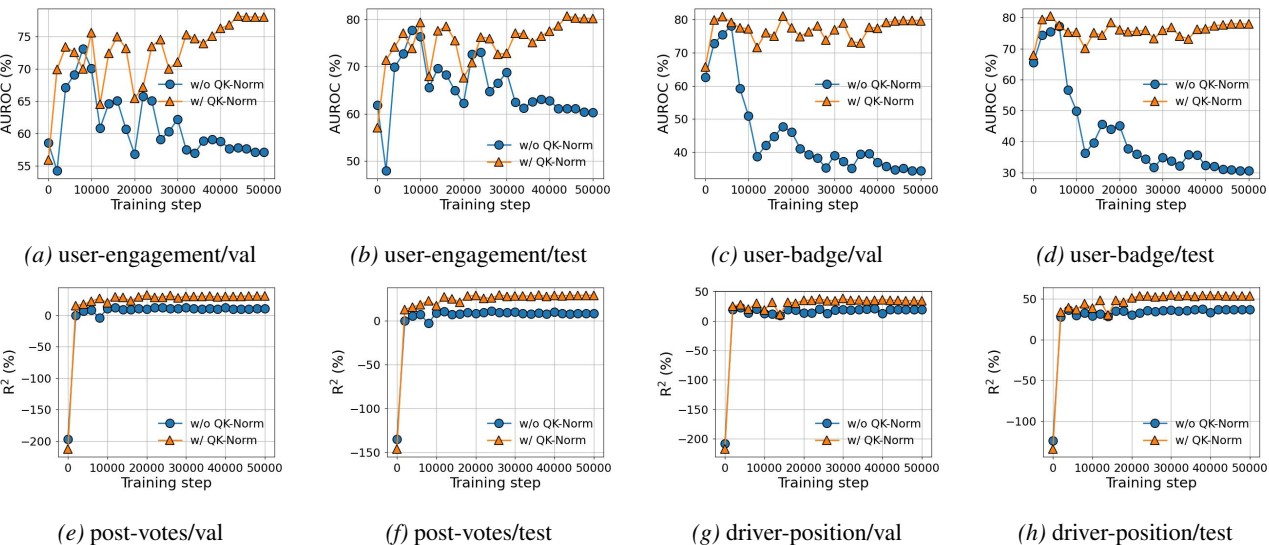

*(a)* user-engagement/val      *(b)* user-engagement/test      *(c)* user-badge/val      *(d)* user-badge/test

*(e)* post-votes/val      *(f)* post-votes/test      *(g)* driver-position/val      *(h)* driver-position/test

*Figure 5.* QK-Norm mitigates early overfitting with *leave-one-db-out* pretraining during the baseline runs and also improves the peak performance. AUROC (%) on the val/test splits of `rel-stack/user-engagement` *(a, b)* and `rel-stack/user-badge` *(c, d)* tasks highlights the mitigation of overfitting. $R^2$(%) on the val/test splits of `rel-stack/post-votes` *(e, f)* and `rel-f1/driver-position` *(g, h)* tasks shows improvements to peak performance.

