# OpenReview forum: "PluRel: Synthetic Data unlocks Scaling Laws for Relational Foundation Models"
_ICML.cc/2026/Conference — ICML 2026 regular_

### Official Review · Reviewer_busc · 2026-03-11

**Soundness:** 3
**Presentation:** 3
**Significance:** 3
**Originality:** 3
**Overall Recommendation:** 4
**Confidence:** 3

**Summary:**

This paper introduced a new framework to synthesize multi-tabular relational databases from scratch, since many helpful relational databases are kept private. The pipeline consists of schema generation, PF_FK modeling, and data filling using SCMs. Experiment results showed that RT's performance increases with diversity and size.

**Compliance With Llm Reviewing Policy:**

Affirmed.

**Final Justification:**

The authors' rebuttal have addressed the concerns. Overall I think it's a paper worth sharing with a broader community, hence I recommend acceptance.

**Key Questions For Authors:**

Please see W1, W2, W3.

**Limitations:**

yes

**Strengths And Weaknesses:**

S1: The paper is well-written and well-motivated. I believe the audience will learn something from the paper.

S2: The overall pipeline makes sense.

W1: There is a rich body of literature on relational data generation in the data management community (e.g. SIGMOD and VLDB). This paper can include a more thorough comparison against those work.

W2: Experiment results can be strengthened. 1) For rel-avito and rel-trial, there's a significant drop in performance, what's the cause for that? 2) It's not clear to me from the paper which tasks are studied in RelBench. These should be explained in better details.

W3: Many important use cases for synthetic, relational data generation goes beyond specific tabular workloads. For example, the data management community is actively studying how to leverage relational data more effectively to do QA. It would be great if the authors can establish a stronger connection between the techniques in this paper, and other, more general LM benchmarks, in addition to just running experiments on RelBench.

---

> ### Author Rebuttal · Authors · 2026-03-31
>
> We appreciate Reviewer busc for pointing us to the literature on data management and the broader tasks with relational QA.
>
> **W1: Regarding related work on synthetic data generation in the data management community**
>
> We noticed that works such as Houkjaer et al. (VLDB 2006) and DataSynth (VLDB 2011) turned out to be relevant to PluRel although the motivation and settings are different. These papers present database engineering tools designed to populate a user-provided schema with structurally correct data. In particular, these approaches fill in cell values independently per column and do not model underlying generative mechanisms that we care about in the machine learning settings. We will add a discussion about them in the related work of the revised paper.
>
> [1] Houkjaer, Kenneth, Kristian Torp, and Rico Wind. "Simple and realistic data generation." Proceedings of the 32nd international conference on Very large data bases. 2006.
>
> [2] Arasu, Arvind, Raghav Kaushik, and Jian Li. "DataSynth: Generating synthetic data using declarative constraints." Proceedings of the VLDB Endowment 4.12 (2011): 1418-1421.
>
> **W2: Regarding perf drop and relbench tasks**
>
> We thank the reviewer for these suggestions. (1) We believe that the performance drops on certain datasets, such as rel-avito and rel-trial, are due to distributional and structural mismatches between the synthetic data and these specific datasets (e.g., differences in schema characteristics, feature distributions, or task difficulty), which can make transfer more challenging. Importantly, these cases are in the minority, and overall, we observe consistent improvements across 7/10 classification and 7/8 regression tasks. (2) We will clearly describe the RelBench datasets and tasks in the revision to improve clarity.
>
> **W3: Regarding synthetic data for relational QA workloads and relevance to LM benchmarks**
>
> We agree that synthetic relational data has applications beyond prediction tasks. In terms of QA with relational data, the text-to-SQL line of work (such as Spider [1] and CoSQL [2]) has been well explored at this point but requires manual annotation of textual queries with their SQL counterparts. This line of work also relies on the semantics of the table and column names. However, PluRel currently does not incorporate semantics into the RDB generation process. Nonetheless, the stage 1 of PluRel can be easily replaced by a custom real world schema (maybe using a large language model or a real-world RDB) with semantically relevant table and column names, followed by PluRel's native stage 2 and stage 3 steps to generate large-scale synthetic RDBs. This essentially allows the diversification of natural language queries that one can convert into corresponding SQL or use directly for QA. Thus, resulting in QA datasets potentially much larger than HybridQA [3] for instance. We will add a discussion on this line of work to extend the scope of PluRel's impact. Thanks for the suggestion.
>
>
> [1] Yu, Tao, et al. "Spider: A large-scale human-labeled dataset for complex and cross-domain semantic parsing and text-to-sql task." Proceedings of the 2018 conference on empirical methods in natural language processing. 2018.
>
> [2] Yu, Tao, et al. "Cosql: A conversational text-to-sql challenge towards cross-domain natural language interfaces to databases." Proceedings of the 2019 conference on empirical methods in natural language processing and the 9th international joint conference on natural language processing (EMNLP-IJCNLP). 2019.
>
> [3] Chen, Wenhu, et al. "HybridQA: A dataset of multi-hop question answering over tabular and textual data." Findings of the Association for Computational Linguistics: EMNLP 2020. 2020.

---

> > ### Author Rebuttal · Reviewer_busc · 2026-04-01
> >
> > Thanks for answering the questions. I stand with my positive view of the paper. Please incorporate the needed discussions in the revised version of the work.

---

> > > ### Author Response · Authors · 2026-04-03
> > >
> > > Thank you for the valuable feedback!

---

### Official Review · Reviewer_Qq8U · 2026-03-13

**Soundness:** 3
**Presentation:** 3
**Significance:** 2
**Originality:** 3
**Overall Recommendation:** 4
**Confidence:** 4

**Summary:**

This paper proposes a synthetic relational database generator for pretraining relational foundation models. The method generates databases in three stages: schema generation with directed graphs, row-level foreign-key connectivity with bipartite graphs, and table features with conditional SCMs. It alsp studies how synthetic database diversity and token count affect relational transformer pretraining, and reports power-law-like loss trends plus gains from synthetic pretraining followed by continued pretraining on real RelBench data.

**Compliance With Llm Reviewing Policy:**

Affirmed.

**Final Justification:**

My final recommendation remains Weak Accept. I think the paper makes a meaningful contribution: the PLUREL framework is well structured, the synthetic relational pretraining setup is interesting, and the continued-pretraining results on real data are encouraging. My main reservations remain about the breadth of the "scaling laws" framing, since the main scaling analysis is still based on held-out synthetic data from the same family, with one model architecture and a limited fitting grid. I also continue to view the lack of direct comparison to alternative relational synthetic generators and the limited ablation of PLUREL’s design choices as important limitations. The rebuttal was helpful and clarified few of my concerns, but my follow-up questions on real-data relevance of the scaling trends and on which generator components matter most were not fully resolved. Overall, I still see the paper as slightly on the positive side of the borderline, so my assessment remains unchanged.

**Key Questions For Authors:**

**1- How much of the claimed scaling behavior survives when evaluated on real rather than synthetic validation distributions?** The current power-law fit is based on held-out PLUREL-generated data. A stronger real-distribution analysis would materially improve my assessment.

2- The power-law fits are performed on a relatively small grid of (N, S) combinations (8 × 7 = 56 points, with some held out). **How stable are the fitted exponents to perturbations in the grid, for instance, if individual runs are excluded? Have the authors checked whether the reported exponents fall within a plausible confidence interval?**

**3- How does PLUREL compare to alternative relational synthetic generators in the same pretraining setting?** Without this, it is difficult to know whether the main contribution is PLUREL specifically or just “more synthetic relational data”

**4- Which PLUREL components matter most?** For example, how sensitive are results to the schema graph prior, HSBM design, SCM family, temporal priors, and random mechanism choices? A convincing ablation would strengthen the soundness substantially.

5- PLUREL does not support self-referential foreign keys (a table referencing itself), which appear in at least one RelBench dataset (rel-stack). **How much of the zero-shot performance gap on rel-stack tasks relative to other datasets is attributable to this structural mismatch, and is this extension planned?**

**6- Why does synthetic-only pretraining transfer poorly, and how can that gap be reduced?** The paper points to missing text and column semantics, a deeper analysis here would significantly improve the practical relevance of the work.

7- The choice of graph priors (BA, RRT, WS) is described but not ablated. **How sensitive is the zero-shot transfer performance to this choice? Would training exclusively on one graph prior for example purely hierarchical RRT graphs degrade performance on flat or hub-centric real databases, and vice versa?**

**8- How broadly should readers generalize the current conclusions?** Are the authors claiming a broad principle for RFMs, or a first strong demonstration for one RT setup on RelBench-style tasks? A tighter scope statement would improve the paper.

**Limitations:**

No. The impact statement is mostly positive and emphasizes privacy and broad enterprise benefits.

**Strengths And Weaknesses:**

**Strengths**

[S1] The paper tackles a real bottleneck for relational foundation models: lack of large, public, diverse multi-table databases. The motivation is good, and the paper identifies an important gap between single-table synthetic generation and relational pretraining data.

[S2] The three-stage decomposition is well structured. Separating schema generation, primary-foreign key connectivity, and feature generation makes the framework easy to understand and gives it a reasonably broad design space.

[S3] The continued-pretraining results on RelBench tasks are good. Synthetic pretraining is not enough on its own, but it does appear to provide a useful initialization for subsequent real-data training, especially on regression tasks.

[S4] The combination of synthetic schema generation, synthetic foreign-key connectivity, and conditional SCM-based generation for RFM pretraining is interesting and different from adapting a standard tabular SCM.

**Weaknesses**

[W1] The scaling law analysis (Section 3.1) fits separate power laws along the N and S axes by marginalizing over the other dimension. The paper explicitly notes that a joint law is inappropriate due to non-monotonicity, but the marginal fitting procedure means the reported exponents characterize best-case scaling rather than typical operating behavior. Practitioners scaling both axes simultaneously, which is the recommended approach, cannot directly use the reported parameters to predict performance.

[W2] The paper’s strongest claim *unlocks scaling laws for RFMs* is too strong for the evidence presented. What is actually shown is a power-law-like fit of validation loss on held-out synthetic databases from the same PLUREL family, with a fixed RT architecture and a small number of frontier points used for fitting. That is interesting, but it is not yet a robust or broad scaling-law result in the stronger sense suggested by the title and abstract.

[W3] The scaling analysis is mostly validated on the generator’s own synthetic distribution. The loss used for fitting is computed on held-out synthetic RDBs, not on real downstream data, so it remains unclear how much of the observed regularity reflects a meaningful law of relational pretraining versus self-consistency of the synthetic family.

[W4] The continued pretraining results (Table 1) show negative absolute gains on three classification tasks. The paper attributes these to semantic sensitivity, but the same tasks also show high variance across seeds (e.g., rel-avito user-clicks: 54.7+-2.9 vs. 47.9+-1.0), making it difficult to determine whether the degradation is systematic or within normal optimization noise.

[W5] The paper does not compare PLUREL against alternative relational synthetic generators in the same pretraining pipeline. The related-work section names graphical-model, GAN, diffusion, and other synthetic relational approaches, but experimentally the paper does not show whether PLUREL is actually better than simpler or prior generators for creating useful RFM pretraining data.

[W6] The method contains many hand-designed priors and modeling choices: graph families, HSBM connectivity, SCM mechanism design, temporal priors, random MLP projections/reconstructions, but the paper provides very little ablation or sensitivity analysis to show which parts matter.

[W7] The generalization evaluation uses only the RelBench benchmark, which comprises 6 databases across a narrow set of domains. It is unclear whether the observed zero-shot improvements reflect genuine relational structure learning or an artifact of domain overlap between synthetic databases and the specific RelBench domains sampled by the chosen graph priors .

[W8] The HSBM-based foreign key connectivity (Section 2.2) imposes a block structure on P->F links that may systematically under-represent the highly skewed, power-law-like degree distributions common in real databases (e.g., popular items with thousands of transactions, inactive users with none). The paper does not discuss whether the chosen connectivity prior can reproduce such distributions, or what effect this mismatch might have on learned relational attention patterns.

[W9] Several important experimental choices are not justified cleanly enough in the main paper, including why the selected synthetic checkpoint is used as the base model, how robust results are to different PLUREL priors, and whether similar conclusions hold for other model sizes or architectures.

[W10] The paper evaluates only a single model architecture (RT, 12 layers) at a single model size. Whether the observed scaling laws hold across model sizes, or whether a different architecture would exhibit different scaling behavior, is not discussed. This limits the degree to which the reported power-law exponents generalize.

---

> ### Author Rebuttal · Authors · 2026-03-31
>
> We appreciate the detailed feedback from Reviewer Qq8U.
>
> **W1 + W2 + W3 + Q1 + Q8**
> - Our goal in Section 3.1 is to isolate the effect of each scaling axis and demonstrate that predictable power-law trends exist when varying one dimension at a time. We agree that the resulting exponents reflect frontier (best-case) behavior rather than joint scaling behavior. At the same time, this marginal analysis is consistent with prior scaling-law work, where frontier fits are used to characterize underlying trends before constructing joint laws. We will clarify this point in the revision.
> - We kindly note that our claim is not that we provide a complete or universal characterization of scaling laws for RFMs, but that we “unlock” scaling laws by demonstrating their existence and empirical relevance in the relational setting. See also our note on future work on joint model and data scaling laws mentioned in Section 5.
> - Please refer to our response to **Reviewer 7ucJ**(addressing their **W1**) for a discussion on scaling analysis on real data. Additionally, please note that our zero-shot and continued pretraining results on RelBench suggest that the observed trends are not purely an artifact of self-consistency.
>
> **W4 + Q6**
>
> Since PluRel does not capture real-world semantics, continued pretraining on real RDBs helps bridge this gap. On a related note, the relational transformer paper [1] also showed that shuffling column and table names harms zero-shot transfer, which highlights the importance of semantics. We will also clarify the potential impact of std dev in the metrics when discussing the results in the revision.
>
> **W5 + Q3**
>
> We note that existing approaches either require access to a real RDB or are not designed to generate diverse RDBs from scratch, making such comparisons non-trivial in our setting. We conducted an additional experiment where we generated synthetic data for rel-f1 using LLM-proposed feature distributions while keeping the underlying schema fixed. We found that such schema-preserving generation did not lead to improvements even in the finetuning settings. Thus, simply generating more data within a fixed schema is insufficient, and that diversity at the level of schemas, foreign key distribution, and SCM mechanisms is critical.
>
> **W6 + Q4 + Q7**
>
> We kindly note that the focus of this work is to study whether synthetic pretraining can improve transfer and exhibit scalable behavior, rather than to fully isolate the contribution of each generator component. However, in our experiments, we noticed that training exclusively on one or two graph priors (such as BA or BA+RRT only) limits data diversity and results in an early onset of overfitting during pretraining. We will include a discussion on these aspects in the revision. More importantly, as also mentioned in our response to Q5 below, the exact schema structure is not necessary for good performance on a particular relbench dataset.
>
> **W7**
>
> Our use of RelBench is motivated by the lack of widely accepted, diverse benchmarks for RFMs. Importantly, our synthetic data generation does not target any specific RelBench domains as the feature generation mechanisms are sampled from general-purpose priors. Therefore, the observed gains are unlikely to arise from direct domain overlap.
>
> **W8**
>
> The HSBM prior does not inherently prevent heavy-tailed degree distributions. In particular, by appropriately choosing the block-level connection probabilities (eq 1) and the relative sizes of blocks, it is possible to induce highly skewed connectivity patterns. We will clarify this point in the paper.
>
> **W9 + W10**
>
> The synthetic checkpoint selection criteria is described in Section 3.3. We also expect that different model sizes and architectures may lead to different scaling regimes or exponent values, as is commonly observed in other domains. We also noted the joint scaling laws as future work in Section 5.
>
> **Q2**
>
> We performed a leave-one-out jackknife over the 7 frontier points used for fitting (N $\in$ {8, …, 512}). The irreducible loss is highly stable across folds (0.357 $\pm$ 0.004), while the power-law exponent $\alpha$ (for fig 1a) showing a variation (0.41 $\pm$ 0.12), which is expected given that we fit 3 parameters to 7 points. However, while the exponent carries uncertainty because of the small grid, the scaling trend itself is robust. We will note this in the revision.
>
> **Q5**
>
> Even without self-referential schema structures in PluRel, we observe +7.4% AUROC on rel-stack/user-engage. While isolating the exact contribution of this mismatch is challenging (as performance also depends on task difficulty), we plan to support such schemas in future work.
>
> [1] Ranjan, Rishabh, et al. "Relational Transformer: Toward Zero-Shot Foundation Models for Relational Data." ICLR 2026

---

> > ### Author Rebuttal · Reviewer_Qq8U · 2026-04-04
> >
> > Thank you for the detailed rebuttal. I appreciate the clarifications provided.
> >
> > - The rebuttal is helpful and addresses several points, but my remaining concerns are mainly about the scope of the strongest claims rather than missing small clarifications. In particular, I still think the evidence supports a strong first demonstration of synthetic relational pretraining and encouraging continued-pretraining results, but not yet a broad characterization of scaling laws for RFMs. The main reason is that the scaling analysis is still centered on held-out synthetic data from the same family, with one model architecture and a limited fitting grid. I also continue to think that the lack of direct comparison to alternative relational synthetic generators and the limited ablation of PLUREL’s design choices make it hard to determine how much of the result is specific to this generator versus the broader idea of synthetic relational pretraining.
> >
> > - On the question of real-data relevance: beyond the zero-shot and continued-pretraining RelBench results, do the authors have any additional evidence that the observed scaling trends are predictive for real validation performance, rather than mainly for held-out synthetic data from the same family?
> >
> > - On the generator design itself: since the current rebuttal suggests that schema diversity is important, do the authors have a clearer sense of which components are most important in practice (schema diversity, FK structure, SCM mechanisms, temporal priors), even if only qualitatively?
> >
> > Overall, the rebuttal is helpful and improves my view of the paper, but these questions still matter for the final interpretation of the contribution. My overall assessment remains unchanged.

---

### Official Review · Reviewer_qXK4 · 2026-03-13

**Soundness:** 3
**Presentation:** 2
**Significance:** 3
**Originality:** 2
**Overall Recommendation:** 4
**Confidence:** 4

**Summary:**

The paper proposed, PLUREL, a framework to synthesize multi-tabular relational databases from scratch. Moreover, the paper provides some analysis on scaling-laws for pre-training and experimental results on Rel-bench to show competitiveness of PLUREL.

**Compliance With Llm Reviewing Policy:**

Affirmed.

**Final Justification:**

The authors have raised many of the concerns during the rebuttal, and with the discussions to be included, it would provide positive insights on generation of synthetic generation of relational data.

**Key Questions For Authors:**

-	How can the synthetic generation be extended for semantics/textual data, which is prevalent for many real-world used cases?
-	How does PLUREL perform with other base models?
-	Why is zero-shot important and what are the reasons for showing only the zero-shot settings? How does PLUREL perform in other settings?
-	With other settings in mind, how does PLUREL (with base models) compare with other RDL or simple baselines?
-	How does the proposed model perform on the extended RelBench with more datasets?
-	What are the computational trade-offs with PLUREL?
-	As the results are presented in percentages, are the standard deviation presented in terms of percentages as well?

**Limitations:**

Yes.

**Strengths And Weaknesses:**

-	In general, the paper is easy to follow.
-	One big challenge of foundation models for relational data is synthetic data generation. The proposed generation of synthetic datasets for relational data can be valuable assets for building RFMs.
-	Experimental comparisons can be improved: it can provide deeper insights with inclusion of baselines (or rather base models) to see where PLUREL can be beneficial on which models.
-	While zero-shot cases are important, it would be interesting to see how PLUREL can benefit on other cases and provide how things can evolve with more known labels.
-	More analysis on computational trade-offs can be helpful.

---

> ### Author Rebuttal · Authors · 2026-03-31
>
> We thank Reviewer qXK4 for the constructive feedback.
>
> **W1 + Q2 + Q4: Regarding base models and RDL baselines**
>
> We kindly note that the field of RFMs is in its early stages, as only the relational transformer (RT) [1] and Griffin [2] models have open-source implementations. We chose RT for our experiments since it outperforms Griffin (as per results in [1]), and Griffin has a limitation that it cannot support auto-complete tasks that we need for synthetic pretraining.
>
> Other baselines such as RDL-GNN are not applicable to zero-shot settings, but we compare against them in the fine-tuning setting below.
>
> **W2 + Q3: Beyond zero-shot settings**
>
> We focus on zero-shot settings as they are important use-cases for RFMs, as argued in [1]. We thank the reviewer for suggesting experiments in other settings. We conducted fine-tuning experiments and find that synthetic pretraining with PluRel followed by continued pretraining on real data shows much higher learning efficiency over real-only pretraining. Please see results below.
>
> _Test R^2 (%), mean over 4 regression tasks:_
> | Fine-Tuning Steps | Synth+Real RT | Real-Only RT | Synth-Only RT | Untrained RT | RDL-GNN  |
>   |------|------------|-----------|------------|------|------|
>   | 16   | 34.4       | 27.7      | 17.6       | 2.7  | 2.1  |
>   | 128  | 38.5       | 37.9      | 30.0       | 12.2 | 5.5  |
>   | 1024 | 40.7       | 40.4      | 35.6       | 28.2 | 11.8 |
>
> **W3 + Q6: Regarding computational tradeoffs**
>
> We provide a detailed analysis of computational trade-offs in Appendix B.4 (Computational Efficiency), including generation time and memory usage of synthetic RDB creation. In particular, we show that PluRel can generate hundreds of databases in parallel with modest CPU resources, making it computationally lightweight relative to downstream model training. We will further clarify and highlight these results in the main text.
>
> **Q1: Regarding incorporating semantics synthetically**
>
> We are currently thinking of the following. A naive way to incorporate text is to collect thousands of words and sample randomly for column names, table names, categoricals etc. However, this approach can backfire and lead to negative transfer based on the column name shuffling experiments in the original RT paper [1]. This approach also fails to use text in the SCM mechanisms, which in turn prevents numeric/text features from being correlated via the SCM mechanisms. One idea is to assign SCM nodes as text dtype and project features to 384-dimensional vectors. These can implicitly act as latent text embeddings (based on what our MiniLM embedding model would generate). So in a way, we are indirectly capturing the embedding of some random text through our SCM mechanisms and using all the 384 dimensions to induce correlations across child nodes. Thanks for raising this point.
>
> **Q5: Regarding results on extended RelBench**
>
> From the extended version of RelBench, we consider the rel-ratebeer dataset with 3 classification and 1 regression task. We take the synthetic pretrained checkpoint used for Table 1 and measure its zero-shot capabilities on this unseen database and tasks.
>
> **zero-shot AUROC (%)**
>
> | Task | Test |
> |------|------|
> | beer-rating-churn | 66.17 |
> | user-rating-churn | 75.49 |
> | brewer-dormant | 70.27 |
>
> **zero-shot R^2 (%)**
>
> | Task | Test |
> |------|------|
> | user-rating-count | 37.75 |
>
> From the results above, we notice that the model achieves strong performance even without any real data pretraining, highlighting the effectiveness of PluRel.
>
> **Q7: Regarding std dev**
>
> Yes. Since the metrics in Table 1 (AUROC and R²) are reported in percentages, the corresponding standard deviations are also expressed in percentage points. We will clarify this in the text to avoid any ambiguity.
>
> [1] Ranjan, Rishabh, et al. "Relational Transformer: Toward Zero-Shot Foundation Models for Relational Data." ICLR 2026.
>
> [2] Wang, Yanbo, et al. "Griffin: Towards a Graph-Centric Relational Database Foundation Model." ICML, 2025.

---

> > ### Author Rebuttal · Reviewer_qXK4 · 2026-04-04
> >
> > Thanks to the authors for the response. The rebuttal has addressed many of my concerns.

---

### Official Review · Reviewer_7ucJ · 2026-03-13

**Soundness:** 4
**Presentation:** 4
**Significance:** 3
**Originality:** 3
**Overall Recommendation:** 5
**Confidence:** 4

**Summary:**

This paper proposes a framework named PluRel for synthesizing relational database tables, aiming to produce large scale data for pre-training relational foundation models (RFMs). PluRel takes three steps: (1) database schema generation through graph sampling; (2) inter-table primary-foreign key connectivity generation with bipartite graphs; and (3) individual table generation with conditional causal mechanisms. Experiments using data synthesized by PluRel to train a Relational Transformer model show that RFMs training loss exhibits power-law scaling with the number of training databases and the number of tokens. Also, pretraining with synthetic data for RFMs enhances subsequent training with real data.

**Compliance With Llm Reviewing Policy:**

Affirmed.

**Final Justification:**

My question on the similarity between the tables in the real dataset and those in the synthetic data has been addressed by the rebuttal. I remain positive about this work.

**Key Questions For Authors:**

There are no further questions.

**Limitations:**

Yes.

**Strengths And Weaknesses:**

Strengths:

1. The paper presents a multi-table relational database synthesis method, while most existing studies consider single table synthesis.

2. This paper provides empirical evidence on scaling laws of relational foundation model pretraining over input data size.

3. The paper is motivated very well and is well written overall.

Weakness:

1. The scaling law observation is made on synthetic data generated based on a few given graph structures. It is not yet clear whether similar observation can be made from real relational database tables.

2. The empirical gain of RFM training using real data and the synthetic data generated by PluRel comparing with using real data only is relatively small (+1.2% and +3.0% in AUROC and R2, respectively, as reported in Table 1). Also, how similar are the tables used in the real dataset experiments to the tables synthesized by PluRel, i.e., are there synthesized tables that are very similar to the real ones?

3. Not a weakness: What does the x-axis of Figure 3 represent?

---

> ### Author Rebuttal · Authors · 2026-03-31
>
> We thank Reviewer 7ucJ for the feedback and comments.
>
> **W1: Regarding scaling laws with real data**
>
> Scaling laws on real-world RDBs are indeed an important direction to explore, but we are bottlenecked by the availability of real RDBs to study the scaling behavior. In particular, PluRel allowed us to generate >1000 synthetic databases to plot the scaling law curves, but in the case of real RDBs, we only have the RelBench collection with < 10 RDBs. Even with the recent release of RelBench v2, the total count is less than 100 RDBs, which is not suitable for scaling law analysis. Nonetheless, the validation loss curves in Figure 4 for unseen real RDBs (in RelBench) indicate that the synthetic scaling insights can indeed be applicable for real data. Furthermore, our goal with the scaling law analysis is to first establish whether a stable and predictable scaling regime exists under a controlled relational data generation setting, analogous to how the early scaling law studies in language modeling relied on curated text corpora (i,e val loss on the same distribution data that is held out).
>
> **W2: Regarding empirical gains and table similarity**
>
> We note that the average improvements (+1.2% AUROC and +3.0% R²) may not fully capture the impact of PluRel, as aggregate metrics can hide differences across tasks (e.g., observe the +7.4% AUROC improvement on rel-stack/user-engage). Through Table 1, we want to show that synthetic pretraining consistently improves transfer to unseen tasks and databases after continued pretraining on real RDBs. In particular, we observe gains on 7/10 classification tasks and 7/8 regression tasks, indicating that PluRel provides a robust, broadly applicable inductive bias rather than task-specific improvements.
>
> In terms of table similarity, when we performed a qualitative comparison with RelBench task tables (e.g., rel-amazon/user-ltv and rel-hm/item-sales), we observed that some synthetic tables can exhibit coarse statistical similarities (such as sparsity or heavy-tailed behavior) to real-world features. For example, the user-ltv column in rel-amazon's train split has quantiles (p25, p50, p75, p95) = (0, 0, 14, 75), while a numeric feature column in one synthetic table exhibits a qualitatively similar but not identical pattern (0, 0, 3, 11). We emphasize that such similarities arise due to the broad distribution of generative mechanisms (such as non-linearities in SCMs), as no real data statistics are used during generation with PluRel.
>
> **W3: Missing caption**
>
> Figure 3 is a collection of histograms, and the x-axis represents the numeric feature values. We will update the caption in the revised manuscript.

---

> > ### Author Rebuttal · Reviewer_7ucJ · 2026-04-03
> >
> > Thank you for the rebuttal. My question on the similarity between the tables in the real dataset and those in the synthetic data has been addressed. I remain positive about this work.

---

### Decision · Program_Chairs · 2026-04-30

**Decision:**

Accept (regular)

**Comment:**

All reviews argue for acceptance.  The paper itself seems like an interesting adaptation of synthetic data to relational foundation models.

Strengths:
1. Method itself (synthetic data for relational FM is novel)
2. Scaling law results
3. The "combination of synthetic schema generation, synthetic foreign-key connectivity, and conditional SCM-based generation for RFM pretraining is interesting and different from adapting a standard tabular SCM."

Most (if not all) weaknesses seemed successfully addressed during the rebuttal phase.